# Where Concept Erasure Should Occur: Concept–Layer Alignment in Text-to-Video Diffusion Models

Yiwei Xie [1]    Ping Liu [* 2]    Zheng Zhang [* 1]

## Abstract

Text-to-video diffusion transformers encode semantic information unevenly across model depth, which constrains effective concept erasure. We identify a representational bottleneck, termed concept–layer topological alignment, under which target concepts exhibit higher separability at certain representational depths. Outside these depths, concept and non-target signals remain strongly entangled, limiting the effectiveness of depth-specific erasure. This observation reframes concept erasure as the problem of identifying representational depths where concept–non-target separation naturally emerges. Motivated by this structural constraint, we introduce CLEAR, a separability-driven optimization framework for concept erasure that explicitly enforces concept–layer alignment. CLEAR operationalizes this principle by formulating layer selection as an optimization problem over concept–non-target separability, rather than relying on layer-agnostic or heuristic choices. To enable this, we introduce a separability-aware objective that favors layers exhibiting stronger concept–non-target separation. Experiments on large-scale text-to-video diffusion models demonstrate that enforcing concept–layer alignment leads to more precise concept suppression while preserving overall generative quality.

## 1. Introduction

Recent text-to-video (T2V) diffusion models exhibit remarkable generative realism (Yang et al., 2025; Kong et al., 2024; Zheng et al., 2024; Wan et al., 2025). However, their in-

*Equal contribution  [1]The School of Artificial Intelligence and Automation, Huazhong University of Science and Technology, Wuhan 430074, China [2]Department of Computer Science and Engineering, University of Nevada, Reno, NV, USA. Correspondence to: Ping Liu <pino.pingliu@gmail.com>, Zheng Zhang <leaf@hust.edu.cn>.

*Proceedings of the 43$^{rd}$ International Conference on Machine Learning*, Seoul, South Korea. PMLR 306, 2026. Copyright 2026 by the author(s).

ternal representations exhibit pronounced depth-dependent structure rather than being uniformly organized across layers (Toker et al., 2024). As semantic concepts are progressively formed and reorganized at different stages of the architecture (Peebles & Xie, 2023; Tian et al., 2025), the effectiveness of concept erasure becomes strongly conditioned on the depth at which it is applied. Consequently, applying the same erasure operation at different layers can lead to markedly different semantic outcomes. This raises a fundamental question for controllable T2V diffusion models: at which representational depth should concept erasure be performed to precisely suppress a desired semantic concept?

Most existing concept erasure and safety-oriented methods implicitly assume that undesired concepts can be suppressed in a layer-agnostic manner, with erasure effects propagating uniformly across the network (Li et al., 2024; Yoon et al., 2025; Ye et al., 2025; Liu & Tan, 2025). Under this assumption, erasure operations are typically applied at fixed or heuristically chosen layers, without explicitly accounting for how semantic representations evolve across depth. However, in T2V diffusion models, semantic representations vary substantially across representational stages, rather than being uniformly organized. As shown in Figure 1, applying erasure at different depths yields distinct outcomes, often suppressing coarse categories while missing fine-grained concepts. This misalignment between erasure placement and representational structure explains the resulting semantic leakage and unintended degradation (Pham et al., 2024; Liu & Zhang, 2025; Zhang et al., 2025).

To better understand this mismatch, we visualize the distribution of positive and negative samples at different layers of the text encoder (Figure 2). This analysis reveals that the effectiveness of concept erasure is constrained by a structural property of deep generative models, which we term concept–layer topological alignment. We define this alignment as the depth-dependent geometric state where the high-dimensional representation of a target concept becomes maximally disentangled from background semantics, forming a linearly isolatable subspace. Specifically, concept separability varies across depth, with different concepts becoming more distinguishable at different depths of the model. This observation aligns with recent findings in representation filtering, which demonstrate that safety concepts

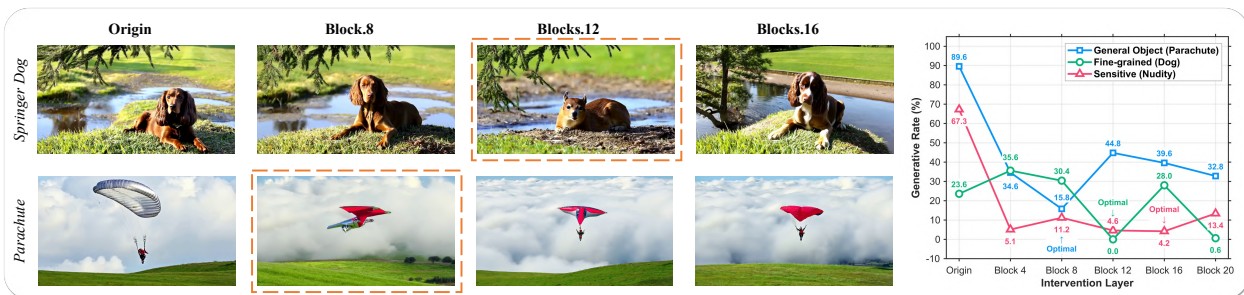

*Figure 1.* **Impact of intervention depth on erasure efficacy.** (Left) The orange dashed boxes indicate optimal intervention layers. Visual results show that deviating from these *topological sweet spots* results in suboptimal erasure or semantic persistence. (Right) The "V-shaped" Generative Rate (percentage of frames in the generated video containing the target concept, lower is better) curves confirm that different concepts are localized at distinct depths, underscoring the need for automated depth discovery over fixed-layer approaches.

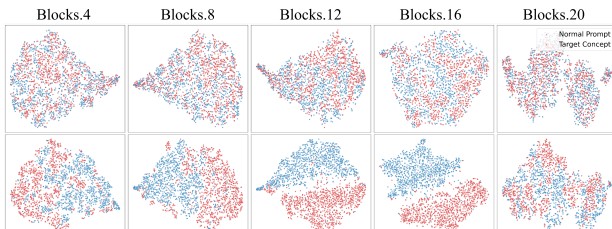

*Figure 2.* **Visualization of Concept-Layer Topological Alignment across Depths.** We compare the layer-wise feature distributions of two distinct concept types using t-SNE. Top: Object concept (e.g., Parachute). Bottom: Sensitive concept (e.g., nudity).

become linearly isolatable only at specific "safety-critical" layers (Li et al., 2025). As a result, effective concept erasure requires intervening at depths where target and non-target semantics are more clearly separated. Identifying such depths, however, is challenging in practice: exhaustive layer-wise exploration is computationally prohibitive, and commonly used selection criteria are not aligned with concept-specific separability. This motivates the need for a principled mechanism for identifying concept-aligned intervention depths.

To address this challenge, we introduce Concept-Layer Erasure Alignment fRamework (CLEAR), a separability-driven optimization framework for concept erasure that directly responds to concept–layer topological alignment. Existing approaches typically assume that erasure placement can be fixed or heuristically chosen, an assumption that conflicts with the depth-dependent emergence of concept separability. CLEAR instead treats the location of erasure as an explicit optimization variable, seeking depths where concept and non-target semantics are most clearly separated. To support this formulation, we introduce a separability-aware objective that prioritizes layers where suppressing concept-specific signals incurs minimal interference with non-target semantics. In this way, CLEAR reframes layer selection from an ad hoc design choice into a principled optimization problem dictated by the model's representational structure.

We evaluate CLEAR on representative large-scale text-to-video diffusion models, including Wan2.2-5B (Wan et al., 2025) and CogVideoX-2B (Yang et al., 2025), across a diverse set of concepts spanning general objects, identities, and safety-sensitive categories. Across all settings, CLEAR consistently outperforms previous state-of-the-art concept erasure methods, achieving substantially stronger suppression while preserving overall generative quality. For example, when erasing french horn from Wan2.2-5B, CLEAR reduces the generative rate from 70.4% to 7.8%, while maintaining comparable imaging and aesthetic quality. In contrast, the previous state-of-the-art method reduces the generative rate only to 26.8% and exhibits more noticeable degradation in visual quality.

In summary, this work makes the following contributions:

- We reveal a depth-dependent representational bottleneck in text-to-video diffusion models, showing that concept erasure effectiveness is governed by concept–layer alignment rather than uniform intervention.

- We propose CLEAR, a concept-dependent framework that automatically selects effective intervention layers by optimizing concept–non-target separability, without updating the diffusion model weights.

- Experiments on large-scale text-to-video diffusion models demonstrate that concept–layer alignment enables more reliable erasure while better preserving non-target semantics and generative quality.

## 2. Related Work

### 2.1. Concept Erasure in Text-to-Video Models

Building upon the success of text-to-image models (Saharia et al., 2022; Rombach et al., 2022), recent T2V systems have rapidly evolved from U-Net-based architectures (Ho et al., 2022; Wu et al., 2023a; Wang et al., 2023) to large-scale diffusion transformers such as CogVideoX (Yang et al.,

2025), HunyuanVideo (Kong et al., 2024), and Wan (Wan et al., 2025). While these models significantly improve realism and temporal coherence, they also inherit the tendency to generate copyrighted, sensitive, or otherwise undesirable content, motivating the need for concept erasure in the video domain. Existing concept erasure methods can be broadly categorized into inference-time and training-time approaches. Inference-time defenses, including negative prompting and safety guidance (Li et al., 2024; Yoon et al., 2025; Xu et al., 2025; Liu & Zhang, 2026), guide the generation process away from target concepts without modifying model parameters, offering a lightweight and flexible solution. However, their effectiveness can depend on prompt structure and may become less stable under complex or compositional prompts. Training-time approaches, such as model fine-tuning or closed-form solutions (Liu & Tan, 2025; Ye et al., 2025; Cheng et al., 2026; Zhang et al., 2026), typically provide stronger and more persistent suppression by directly modifying model behavior. Nevertheless, because these methods intervene at the parameter level, they may introduce non-target changes when the target concept is entangled with related semantics. Despite their differences, these methods largely assume that concept erasure can be applied in a layer-agnostic or heuristically fixed manner, an assumption that overlooks the depth-dependent organization of semantic representations in diffusion transformers and becomes particularly limiting in text-to-video models.

## 2.2. Mechanistic Interpretability

Mechanistic interpretability aims to uncover the internal organization of neural networks by decomposing dense activations into more interpretable components (Pham et al., 2026; Yan et al., 2025; Bussmann et al., 2025). A prominent line of work in this area focuses on sparse representation learning (Zhang et al., 2015; He et al., 2025b; Thasarathan et al., 2025), which has been shown to disentangle polysemantic activations into more semantically coherent features in large language models and diffusion-based generative models (He et al., 2026; Tian et al., 2025). In the text-to-image domain, recent works have successfully repurposed SAEs for precise concept erasure, ranging from zero-shot feature deactivation (Tian et al., 2025; Cywiński & Deja, 2025) to supervised neuron localization (Cassano et al., 2025; He et al., 2025a). While these approaches achieve impressive precision by operating in the sparse feature space, they largely focus on which features to suppress, while treating the intervention depth as a fixed hyperparameter or relying on manual selection layers. The depth-dependent emergence and separability of semantic features remain underexplored, limiting the applicability of these methods to settings where concept representations vary substantially across model depth.

## 3. Method

### 3.1. Problem Formulation

We study concept erasure in diffusion transformers through two tightly coupled questions: *where* to intervene within the model's depth hierarchy, and *how* to perform the intervention once a suitable depth is identified.

Given a pre-trained text-to-video diffusion model $\epsilon_\theta(\mathbf{z}_t, t, c)$ and a target concept $C$ encoded in its parameters, we consider an erasure operation $\mathcal{E}$ applied to an intermediate representation. Let $\mathbf{h}_l$ denote the activation at layer $l$; intervention is modeled as

$$\mathbf{h}'_l = \mathcal{E}(\mathbf{h}_l).$$

Applying the same operation at different depths generally leads to different semantic effects, as concept and non-target information are encoded in a depth-dependent manner.

Our approach treats intervention depth as a learnable and differentiable variable, allowing the model to identify layers where concept–non-target separability is most pronounced. Conditioned on the selected depth, we then perform feature-level intervention to selectively suppress concept-related signals while preserving non-target semantics. Together, these components form a unified framework that aligns intervention location with representation-level control for effective concept erasure.

### 3.2. Learning Where to Erase: Continuous Relaxation of Intervention Depth

The analysis in the previous section shows that concept–non-target separability varies substantially across model depth. As a result, the effectiveness of concept erasure depends critically on where the intervention is applied. However, manually selecting an intervention layer is impractical for large diffusion transformers and diverse target concepts, and fixed-layer choices do not generalize across concepts or models. We therefore formulate intervention depth as a learnable variable and optimize it directly.

**Depth-wise Preference Modeling.** Rather than committing to a single intervention layer *a priori*, we model intervention depth as a soft preference distribution over all layers. This allows representational evidence to guide where erasure should occur.

Let $\boldsymbol{\alpha} = \{\alpha_1, \alpha_2, \ldots, \alpha_L\}$ denote learnable depth preference parameters, where each $\alpha_l$ reflects the relative suitability of intervening at layer $l$. This assigns a continuous preference over the model's representational hierarchy instead of enforcing a hard layer choice. We construct a unified framework in which the SAE-based feature intervention can, in principle, be applied at any layer, while the contribution of each layer is modulated by its corresponding preference weight. This enables gradual and data-driven

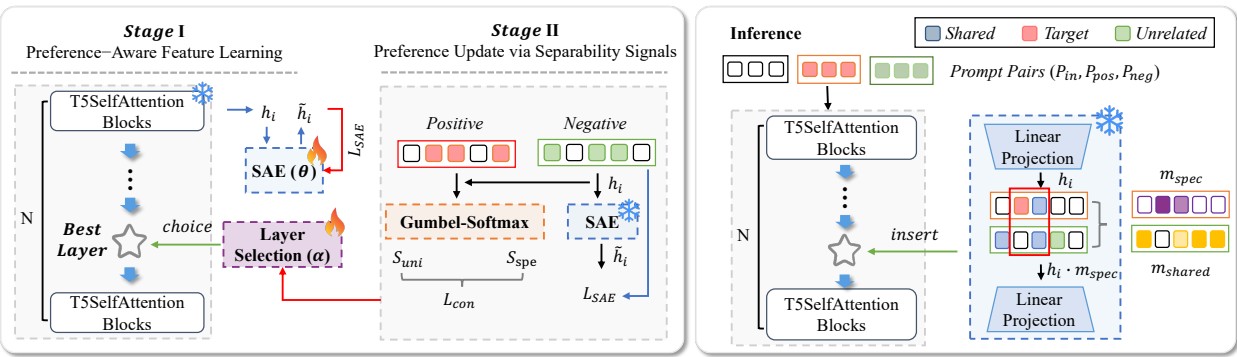

*Figure 3.* **The pipeline of CLEAR.** During training, we utilize discrete positive/negative prompt pairs and Gumbel-Softmax relaxation to differentiably search for the layer index $N$ that maximizes concept separability. The framework trains a SAE to decompose the hidden states, using a contrastive loss ($L_{con}$) to isolate the target concept direction ($W_{spe}$) from universal features ($W_{uni}$). During inference, the learned SAE is inserted into the frozen T5 encoder at the identified optimal layer. The target concept is erased by subtracting the aligned sparse feature vector ($v_{tar}$) from the hidden state $h_i$, preventing semantic leakage into the generation process.

refinement of intervention depth, rather than reliance on heuristic or fixed selection.

**Differentiable Depth Sampling.** Optimizing over discrete layer indices is non-differentiable and incompatible with gradient-based learning. To enable end-to-end optimization of depth preferences, we employ a differentiable sampling mechanism based on the Gumbel–Softmax relaxation.

During training, this mechanism produces a soft, approximately one-hot vector $\mathbf{p} \in \mathbb{R}^L$ that represents the current preference over intervention layers. The $k$-th element of $\mathbf{p}$ is computed as

$$\mathbf{p}_k = \frac{\exp((\log \alpha_k + g_k)/\tau)}{\sum_{j=1}^{L} \exp((\log \alpha_j + g_j)/\tau)}, \qquad (1)$$

where $g_k \sim \mathrm{Gumbel}(0, 1)$ are independent noise samples and $\tau$ controls the sharpness of the distribution. As the temperature $\tau$ anneals during training, the soft preference distribution ultimately converges to a single, discrete layer index for the intervention.

### 3.3. Learning How to Erase: Concept-Aligned Representational Directions

Identifying a favorable intervention depth determines where concept erasure is feasible, but does not specify how erasure should be performed. Even at such depths, directly modifying internal activations often suppresses the target concept while degrading unrelated semantics. This arises from strong representational entanglement in diffusion transformers, where individual latent dimensions encode multiple semantic factors (Wu et al., 2023b; 2025). Effective concept erasure therefore requires operating on representations that support selective attenuation of the target concept.

**Concept-Aligned Feature Decomposition.** To enable selective intervention, we transform dense activations into a feature space with improved semantic coherence. Dense representations $\mathbf{h} \in \mathbb{R}^{B \times T \times d_{model}}$ superpose multiple attributes, making direct manipulation unreliable. We therefore adopt Sparse Autoencoders (SAEs) (Huben et al., 2024) to decompose activations into sparse semantic features.

Given an intermediate activation $\mathbf{h}$, a shared SAE projects it into a higher-dimensional sparse feature space $\mathbb{R}^{d_{sae}}$ (with $d_{sae} \gg d_{model}$):

$$\mathbf{f} = \mathrm{ReLU}(\mathbf{W}_{enc}\mathbf{h} + \mathbf{b}_{enc}). \qquad (2)$$

This representation provides a structured interface for manipulating concept-related components while largely preserving other semantics.

**Separating Concept and Non-Target Semantics.** Sparse features activated by the target concept are not necessarily exclusive to it; many also support non-target semantics. To distinguish these roles, we compare feature activations between a *Positive Set* of prompts containing the target concept $C$ and a *Negative Set* of related prompts that exclude $C$. We capture this distinction using continuous feature weights. Specifically, we define a *Shared Mask* $\mathbf{m}_{shared} \in [0,1]^{d_{sae}}$ based on feature activations over the negative set:

$$\mathbf{m}_{shared}^{(i)} = \frac{f_{neg}^{(i)}}{\max_k(f_{neg}^{(k)}) + \epsilon}, \qquad (3)$$

where larger values indicate stronger association with non-target semantics. We then define a complementary *Specificity Mask*

$$\mathbf{m}_{spec}^{(i)} = 1 - \mathbf{m}_{shared}^{(i)}, \qquad (4)$$

which emphasizes features more aligned with the target concept.

**From Disentanglement to Search Guidance.** Beyond enabling inference-time erasure, these masks serve a critical role in our training framework: providing a quantitative metric for topological search. Because the raw separability varies across layers (Sclocchi et al., 2025), the contrast between the concept-specific energy (filtered by $\mathbf{m}_{spec}$) and the shared energy (filtered by $\mathbf{m}_{shared}$) acts as a robust indicator of the "topological sweet spot." This allows us to transform the qualitative notion of "entanglement" into a differentiable learning signal ($\mathcal{L}_{con}$), which we utilize in the subsequent optimization stage to steer the depth preferences $\boldsymbol{\alpha}$ toward the optimal layer.

### 3.4. Training via Separability-Driven Optimization

Our formulation introduces two coupled sets of learnable parameters: (i) depth preferences $\boldsymbol{\alpha}$ that determine *where* to apply concept erasure, and (ii) the parameters $\boldsymbol{\theta}$ that govern the feature representations produced by the SAE, which determine *how* erasure is realized at a given depth. These two components are intrinsically coupled: depth preferences affect which layers are emphasized during training, while the quality of separability estimates for updating $\boldsymbol{\alpha}$ depends on the current feature decomposition.

A key challenge is that reconstruction fidelity alone is not aligned with the objective of concept erasure. A layer that preserves general information well may still exhibit poor separation between target and non-target semantics, causing depth optimization to favor representations that are stable but semantically entangled. Therefore, effective depth selection requires a training signal that reflects concept–non-target separability rather than reconstruction quality alone. To this end, we optimize the depth preferences $\boldsymbol{\alpha}$ and the SAE parameters through an alternating procedure, allowing the layer-selection component (*where*) and the feature-decomposition component (*how*) to co-evolve. This coupling ensures that depth preferences are updated using separability estimates from the current SAE, while the SAE is trained on representations emphasized by the current depth distribution. We instantiate this coupled optimization through two alternating steps, each updating one component while using the other as a fixed reference.

*Step 1: Preference-Aware Feature Learning.* With the current depth preference distribution $\mathbf{p}$ fixed, the SAE is trained to minimize a weighted reconstruction objective:

$$\mathcal{L}_{SAE}(\theta) = \mathbb{E}_{\mathbf{x} \sim \mathcal{D}} \left[ \left\| \mathbf{h}_l - \sum_{l=0}^{L-1} p_l \cdot \hat{\mathbf{h}}_l \right\|_2^2 + \lambda \|\mathbf{f}_l\|_1 \right]. \quad (5)$$

This step encourages the SAE to allocate representational capacity toward layers that are currently favored by $\mathbf{p}$, so that separability estimates used for depth selection are computed in a consistent feature space.

*Step 2: Preference Update via Separability Signals.* With SAE parameters fixed, the depth preference parameters $\boldsymbol{\alpha}$ are updated by minimizing the combined objective $\mathcal{L}_{CLEAR} = \mathcal{L}_{con} + \mathcal{L}_{SAE}$. Let $S = \sum(\mathbf{f} \odot \mathbf{m})$ denote an aggregate activation score over a prompt set. We formulate $\mathcal{L}_{con}$ to penalize the dominance of shared, non-target energy ($S_{uni}$) over concept-specific energy ($S_{spe}$):

$$\mathcal{L}_{con}(\alpha) = \log \left( 1 + \frac{S_{uni}}{S_{spe} + \epsilon} \right). \quad (6)$$

where $\epsilon$ is a small constant (e.g., $10^{-8}$) added for numerical stability to avoid division by zero when $S_{spe}$ approaches zero. In this step, separability signals derived from the SAE guide the preference distribution toward layers that consistently exhibit clearer concept–non-target separation. Through repeated alternation, the depth preference distribution sharpens around layers that are more suitable for effective concept erasure.

### 3.5. Inference-Time Intervention

After optimization, the continuous architecture distribution collapses to a deterministic choice $l^* = \arg\max_l \alpha_l$. At inference time, we attach the trained SAE $\theta^*$ only at this selected layer and perform concept erasure without modifying the model parameters. The selected layer determines where the target concept is most separable; the remaining step is therefore to remove the concept-specific component from the activation at this layer.

Given a hidden activation $\mathbf{h}_{l^*}$, we first obtain its sparse representation $\mathbf{f}$ through the SAE encoder. Using the specificity mask $\mathbf{m}_{spec}$, we estimate the component of the activation that corresponds to the target concept and project it back to the model space:

$$\begin{aligned} \mathbf{v}_{tar} &= \mathbf{W}_{dec}(\mathbf{f} \odot \mathbf{m}_{spec}), \\ \mathbf{h}'_{l^*} &= \mathbf{h}_{l^*} - \gamma \, \mathbf{v}_{tar}, \end{aligned} \quad (7)$$

where $\gamma$ controls the intervention strength. This operation selectively attenuates target-concept directions while preserving the remaining representation. [1]

## 4. Experiments

### 4.1. Experimental Setup

**Models and Baselines.** We evaluate CLEAR on two state-of-the-art text-to-video diffusion transformers, Wan2.2-5B (Wan et al., 2025) and CogVideoX-2B (Yang et al., 2025), covering different model scales. Interventions are applied to blocks in the T5-based text encoder. We

---

[1]As the intervention is applied only at inference time and localized to a single layer in the trained SAE, it does not require modifying the diffusion model's weights.

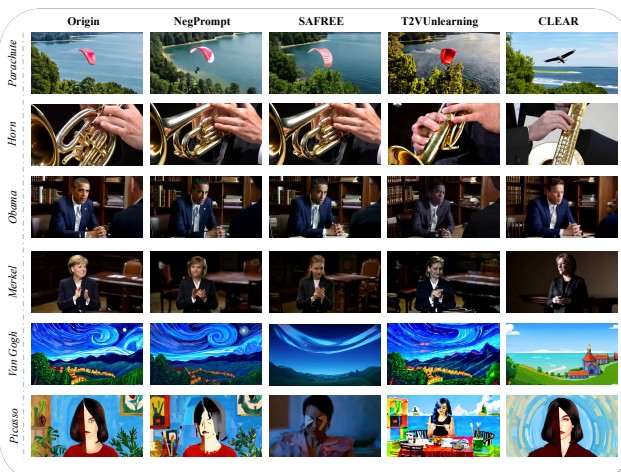

*Figure 4.* **Qualitative comparison of concept erasure.** We present visual results of CLEAR alongside three state-of-the-art baselines: NegPrompt, SAFREE, and T2VUnlearning.

compare against three representative baselines: Neg-Prompt (Li et al., 2024), SAFREE (Yoon et al., 2025), and T2VUnlearning (Ye et al., 2025).

**Erasure Concepts.** To evaluate the robustness of CLEAR across different semantic granularities, we consider four categories of target concepts: (1) general semantic objects, including ten common object categories such as parachute and springer dog; (2) safety-regulated concepts, focusing on nudity as a representative target in content moderation for open-world T2V systems; (3) identity-specific concepts, including celebrity identities, used to assess whether a specific identity can be removed without degrading the generation quality of non-target human figures; and (4) artist style concepts, covering distinctive artistic styles (e.g., Van Gogh), which serve to verify the method's capability to erase abstract stylistic attributes while preserving the underlying semantic content structure. For each concept, we use LLM to generate positive/negative prompt pairs (e.g. "a stunt performer with a parachute over a desert landscape, motion blur, backlit", vs. "a stunt performer over a desert landscape, motion blur, backlit") to guide the SAE in identifying and isolating the specific concept direction during training. With the exception of the artist style, all other concept videos were generated using a fixed seed of 42.

**Evaluation Protocol.** We adopt four complementary metrics to evaluate erasure performance and generation quality. (1) Erasure effectiveness is measured by the post-erasure generation rate, defined as the percentage of generated frames that still contain the target concept, detected using pre-trained concept classifiers (e.g., NudeNet (Bedapudi, 2022) for nudity and FaceNet (Deng et al., 2022) for celebrities). (2) Overall consistency is assessed using ViCLIP similarity (Wang et al., 2024) between the generated video

and the original prompt, evaluating semantic preservation of non-target content. (3) Visual fidelity is measured using the MUSIQ image quality predictor (Ke et al., 2021) trained on the SPAQ dataset, computed frame-wise. (4) Aesthetic quality is evaluated using an aesthetic predictor (e.g., LAION-Aesthetic) (LAION-AI, 2022) to estimate human-perceived visual appeal.

### 4.2. Results on Object Concepts

We evaluate CLEAR on ten general object categories across two T2V architectures, Wan2.2-5B and CogVideoX-2B. Table 1 shows that CLEAR maintains effective concept erasure while preserving video quality across both settings.

On Wan2.2-5B, CLEAR achieves the strongest overall performance for object concept erasure. It reduces the average Generative Rate from 61.8% to 12.8%, substantially outperforming the training-based baseline T2VUnlearning (24.5%) as well as inference-time methods such as Neg-Prompt (54.3%) and SAFREE (28.1%). At the same time, CLEAR preserves visual fidelity: while baseline methods typically degrade image quality (e.g., T2VUnlearning reduces Imaging Quality to 0.6652), CLEAR maintains and slightly improves it (0.7025 compared to 0.6910 for the original model). This indicates that selectively attenuating concept-aligned SAE features removes target-related artifacts without disrupting the model's overall distribution. Moreover, CLEAR achieves an Overall Consistency score of 0.1896, comparable to T2VUnlearning (0.1905), showing that it matches the semantic suppression strength of global fine-tuning while avoiding its adverse impact on quality.

On CogVideoX-2B, the results further demonstrate the stability and precision of CLEAR. Although the parameter-update baseline T2VUnlearning reduces the concept frequency to 7.4%, it severely damages video fidelity, with Imaging Quality dropping sharply to 0.3907, indicating that global weight modification on smaller architectures can induce catastrophic forgetting. In contrast, CLEAR achieves the lowest Generative Rate at 7.1%, outperforming all baselines while preserving substantially higher Imaging Quality (0.4683) and Aesthetic Quality (0.4484). Across diverse models and erasure paradigms, Motion Smoothness remains largely comparable to the original models, suggesting that CLEAR does not substantially disrupt temporal continuity. These results show that operating in a disentangled sparse feature space, rather than modifying dense model parameters, allows CLEAR to decouple concept erasure from generation quality, yielding a more robust and reliable solution for concept erasure.

To provide a qualitative perspective, Figure 4 visualizes erasure results for the parachute concept. Existing methods exhibit a common failure mode: all three baselines tend to perform lazy erasure, where the target remains largely intact,

*Table 1.* **Quantitative comparison of object concept erasure across two architectures.** We compare CLEAR with NegPrompt, SAFREE, and T2VUnlearning on erasure effectiveness, imaging quality, and aesthetic quality. CLEAR consistently achieves stronger concept suppression while maintaining competitive visual fidelity.

*(a)* Performance on Wan2.2-5B.

| Erasure Method | Cassette player | Chain saw | Church | Springer dog | French horn | Garbage truck | Gas pump | Golf ball | Parachute | Tench | Average ↓ | Ove. ↑ | Img. ↑ | Aes. ↑ | Mot. ↑ |
|---|---|---|---|---|---|---|---|---|---|---|---|---|---|---|---|
| Origin | 26.0 | 74.2 | 64.6 | 23.6 | 70.4 | 85.0 | 79.8 | 100.0 | 89.6 | 4.8 | 61.8 | 0.2490 | 0.6910 | 0.5981 | 0.9886 |
| NegPrompt | 14.6 | 57.2 | 50.2 | 30.6 | 43.2 | 76.4 | 81.2 | 90.0 | 85.6 | 13.8 | 54.3 | **0.2556** | 0.6644 | **0.5882** | 0.9852 |
| SAFREE | 13.0 | 11.6 | 24.4 | 16.8 | 28.6 | 32.6 | 40.0 | 63.2 | 49.4 | 1.2 | 28.1 | 0.2212 | 0.6304 | 0.5557 | 0.9888 |
| T2VUnlearning | 3.2 | 27.8 | 33.0 | 15.2 | 26.4 | 7.4 | 71.8 | 0.8 | 57.0 | 2.6 | 24.5 | 0.1905 | 0.6652 | 0.5197 | 0.9848 |
| **CLEAR** | 0.0 | 9.2 | 27.2 | 4.8 | 7.8 | 11.6 | 31.6 | 14.2 | 18.4 | 3.6 | **12.8** | 0.1896 | **0.7025** | 0.5758 | 0.9887 |

*(b)* Performance on CogVideoX-2B.

| Erasure Method | Cassette player | Chain saw | Church | Springer dog | French horn | Garbage Truck | Gas pump | Golf ball | Parachute | Tench | Average ↓ | Ove. ↑ | Img. ↑ | Aes. ↑ | Mot. ↑ |
|---|---|---|---|---|---|---|---|---|---|---|---|---|---|---|---|
| Origin | 9.4 | 42.4 | 59.7 | 10.6 | 80.3 | 78.8 | 79.1 | 96.5 | 69.4 | 36.2 | 56.2 | 0.2447 | 0.5505 | 0.5100 | 0.9774 |
| NegPrompt | 1.8 | 18.2 | 24.1 | 11.5 | 24.4 | 19.4 | 21.2 | 88.8 | 33.2 | 26.2 | 26.9 | **0.2372** | **0.5106** | **0.4878** | 0.9761 |
| SAFREE | 0.6 | 5.6 | 3.5 | 0.0 | 7.4 | 22.7 | 34.7 | 46.8 | 24.1 | 0.3 | 14.6 | 0.1924 | 0.4522 | 0.4399 | 0.9763 |
| T2VUnlearning | 0.0 | 0.0 | 19.1 | 5.6 | 1.5 | 0.0 | 5.9 | 14.4 | 8.8 | 19.1 | 7.4 | 0.1751 | 0.3907 | 0.4082 | 0.9545 |
| **CLEAR** | 0.0 | 0.9 | 23.8 | 0.0 | 0.0 | 11.5 | 10.6 | 22.7 | 10.6 | 1.5 | **7.1** | 0.1747 | 0.4683 | 0.4484 | 0.9778 |

*Table 2.* **Quantitative evaluation of nudity and celebrity concept erasure on Wan2.2-5B.** For celebrity erasure, we report the Identity Similarity Score (lower is better) for five distinct public figures.

*(a)* Nudity concept erasure.

| Erasure Method | Generative Rate ↓ | Overall Consistency ↑ | Imaging Quality ↑ | Aesthetic Quality ↑ | Motion Smoothness ↑ |
|---|---|---|---|---|---|
| Origin | 67.3 % | 0.2312 | 0.6913 | 0.5552 | 0.9953 |
| NegPrompt | 55.5 % | **0.2366** | 0.6453 | **0.5639** | 0.9945 |
| SAFREE | 48.7 % | 0.2169 | 0.6248 | 0.5387 | 0.9944 |
| T2VUnlearning | 18.6 % | 0.2168 | 0.6737 | 0.5322 | 0.9937 |
| CLEAR | **11.1 %** | 0.1754 | **0.6928** | 0.5546 | **0.9951** |

*(b)* Celebrity concept erasure.

| Erasure Method | Merkel | Obama | Trump | Biden | Elizabeth | Average |
|---|---|---|---|---|---|---|
| Origin | 0.1519 | 0.6589 | 0.5652 | 0.5764 | 0.5252 | 0.4955 |
| NegPrompt | 0.0219 | 0.4500 | 0.2590 | 0.2646 | 0.2667 | 0.2524 |
| SAFREE | 0.0058 | 0.1266 | 0.0158 | 0.1117 | 0.2901 | 0.1100 |
| T2VUnlearning | **-0.0551** | 0.2447 | **-0.0506** | **-0.0382** | 0.0068 | 0.0215 |
| CLEAR | -0.0204 | **0.0651** | 0.0812 | 0.0301 | **-0.0803** | **0.0151** |

indicating insufficient or overly conservative intervention. More qualitative results are provided in Appendix F.

### 4.3. Results on Nudity Concepts

We further evaluate CLEAR on the erasure of nudity, a highly sensitive concept that is deeply entangled with human structure and therefore difficult to suppress without degrading general human generation. Quantitative results on Wan2.2-5B and CogVideoX-2B are reported in Table 2a and Table 3a. Across both architectures, CLEAR consistently achieves the strongest erasure performance. On Wan2.2-5B (Table 2a), CLEAR attains the lowest Generative Rate of 11.1%, substantially outperforming the training-based baseline T2VUnlearning (18.6%). In contrast, inference-time methods struggle with this deeply embedded semantic concept: NegPrompt and SAFREE leave high residual nudity rates of 55.5% and 48.7%, respectively. These results highlight the limitation of superficial prompt-level guidance for safety-critical concepts and demonstrate CLEAR's ability to surgically suppress nudity while preserving human realism.

### 4.4. Results on Celebrity Concepts

We evaluate CLEAR on celebrity identity erasure, a challenging setting that requires precise removal of identity-specific features without degrading general human gen-

eration. As reported in Table 2b and Table 3b, CLEAR consistently achieves the lowest Identity Similarity Scores across all celebrities on both architectures. On Wan2.2-5B, CLEAR reduces the similarity score for Obama from 0.6589 to 0.0651, substantially outperforming T2VUnlearning (0.2447) and SAFREE (0.1266), and further drives the scores for Merkel and Elizabeth into negative values, indicating that generation is actively steered away from the target identity manifold rather than merely suppressed. Similar trends are observed on CogVideoX-2B, where inference-time methods such as NegPrompt retain high identity similarity (e.g., 0.4312 for Obama), while CLEAR achieves consistent reduction and outperforms T2VUnlearning on multiple identities, including Biden and Elizabeth. These results demonstrate that surgical intervention in sparse feature space enables robust and precise identity-level erasure without compromising general human generation. Furthermore, we compare CLEAR with VideoEraser, a recent inference-time baseline. Although VideoEraser obtains a lower average identity similarity score on CogVideoX-2B, this improvement comes with online gradient-based optimization during inference, resulting in approximately $1.4\times$ computational overhead. In contrast, CLEAR performs erasure through a cached SAE-based intervention at the selected layer, introducing negligible additional latency once the target concept has been optimized. Moreover, CLEAR remains competitive in identity suppression and outperforms Video-

*Table 3.* **Quantitative comparison of nuidity and celebrity erasure on CogVideoX-2B.** For celebrity erasure, we report the Identity Similarity Score (lower is better) for five distinct public figures.

*(a)* Nudity concept erasure.

| Erasure Method | Generative Rate ↓ | Overall Consistency ↑ | Imaging Quality ↑ | Aesthetic Quality ↑ | Motion Smoothness ↑ |
|---|---|---|---|---|---|
| Origin | 56.14 % | 0.2209 | 0.4671 | 0.4595 | 0.9918 |
| NegPrompt | 42.82 % | **0.2224** | **0.4765** | **0.4812** | 0.9921 |
| SAFREE | 35.16 % | 0.2164 | 0.4462 | 0.4592 | 0.9907 |
| T2VUnlearning | 19.63 % | 0.2058 | 0.3795 | 0.4235 | 0.9916 |
| VideoEraser | 19.22 % | 0.2101 | 0.4530 | 0.4858 | **0.9946** |
| CLEAR | **14.63 %** | 0.2008 | 0.4592 | 0.4467 | 0.9894 |

*(b)* Celebrity concept erasure.

| Erasure Method | Merkel | Obama | Trump | Biden | Elizabeth | Average |
|---|---|---|---|---|---|---|
| Origin | 0.1816 | 0.4653 | 0.6503 | 0.5627 | 0.3222 | 0.4643 |
| NegPrompt | 0.1329 | 0.4312 | 0.4267 | 0.4148 | 0.3011 | 0.3413 |
| SAFREE | 0.1048 | 0.2969 | 0.4448 | 0.3366 | 0.3515 | 0.3069 |
| T2VUnlearning | **0.0270** | 0.2684 | **0.2621** | 0.2290 | 0.2537 | 0.2080 |
| VideoEraser | 0.1430 | 0.1648 | -0.0207 | 0.1921 | 0.0907 | 0.1140 |
| CLEAR | 0.0819 | **0.2396** | 0.3325 | **0.1874** | **0.0594** | **0.1802** |

*Table 4.* **Quantitative evaluation of artist style erasure on Wan2.2-5B.** We report the Video-CLIP score for *Style Erasure* ($VCLIP_e$, lower is better) and *Semantic Preservation* ($VCLIP_s$, higher is better). $H_a$ denotes the *Trade-off Score* ($H_a = VCLIP_s - VCLIP_e$), which quantifies the net effectiveness of the intervention.

| Erasure Method | Pablo Picasso | | | Van Gogh | | | Rembrandt | | | Andy Warhol | | | Caravaggio | | |
|---|---|---|---|---|---|---|---|---|---|---|---|---|---|---|---|
| | VCLIP$_e$ ↓ | ViCLIP$_s$ ↑ | H$_a$ ↑ | VCLIP$_e$ ↓ | VCLIP$_s$ ↑ | H$_a$ ↑ | VCLIP$_e$ ↓ | VCLIP$_s$ ↑ | H$_a$ ↑ | VCLIP$_e$ ↓ | VCLIP$_s$ ↑ | H$_a$ ↑ | VCLIP$_e$ ↓ | VCLIP$_s$ ↑ | H$_a$ ↑ |
| Origin | 0.1804 | 0.1837 | / | 0.2388 | 0.1691 | / | 0.1566 | 0.1897 | / | 0.1450 | 0.1926 | / | 0.1945 | 0.1802 | / |
| NegPrompt | 0.2270 | **0.2189** | -0.0081 | 0.2483 | **0.1892** | -0.0591 | 0.1896 | **0.2159** | 0.0263 | 0.1559 | **0.2242** | 0.0683 | 0.1882 | **0.2190** | 0.0308 |
| SAFREE | 0.1494 | 0.1353 | -0.0141 | 0.1304 | 0.1036 | -0.0268 | **0.0588** | 0.1319 | 0.0731 | **0.0535** | 0.1442 | 0.0907 | 0.1172 | 0.1119 | 0.0018 |
| T2VUnlearning | 0.1447 | 0.1325 | -0.0122 | 0.2137 | 0.1170 | -0.0967 | 0.0630 | 0.1590 | 0.0906 | 0.0717 | 0.1671 | 0.0954 | 0.1314 | 0.1282 | -0.0032 |
| CLEAR | **0.1397** | 0.1664 | **0.0267** | **0.1074** | 0.1521 | **0.0447** | 0.0593 | 0.1901 | **0.1308** | 0.0669 | 0.1798 | **0.1129** | 0.1057 | 0.1543 | **0.0486** |

*Table 5.* **Ablation Study on Search Strategy and Objective Function across Diverse Concepts.** We compare the selected intervention layer and erasure performance for *Springer Dog*, *Parachute*, and *Nudity*. "Manual" denotes the optimal layer identified via brute-force search (High Cost $N_{Layer}\times$). "w/o $\mathcal{L}_{con}$" denotes CLEAR guided solely by reconstruction loss.

| Concept | Method | Selected Layer (Block ID) | Generative Rate ↓ | Imaging Quality ↑ | Aesthetic Quality ↑ | Search Cost |
|---|---|---|---|---|---|---|
| Springer Dog | Manual | 12 | 0.0% | 0.6218 | 0.5989 | High |
| | w/o $\mathcal{L}_{con}$ | 15 | 14.0% | 0.7460 | 0.5599 | Low |
| | w/ $\mathcal{L}_{con}$ | **2** | **4.8%** | **0.6421** | **0.6048** | **Low** |
| Parachute | Manual | 8 | 15.8% | 0.7212 | 0.5814 | High |
| | w/o $\mathcal{L}_{con}$ | 13 | 27.0% | 0.7076 | 0.6080 | Low |
| | w/ $\mathcal{L}_{con}$ | **6** | **18.4%** | **0.7255** | **0.6010** | **Low** |
| Nudity | Manual | 16 | 4.2% | 0.5614 | 0.4491 | High |
| | w/o $\mathcal{L}_{con}$ | 14 | 9.4% | 0.6292 | 0.5196 | Low |
| | w/ $\mathcal{L}_{con}$ | **18** | **10.9%** | **0.6806** | **0.4901** | **Low** |

*Table 6.* **Ablation on Top-$k$ Layer Selection and $\gamma$ with Wan2.2-5B on Nudity.**

| Intervention | Gen. Rate | Consistency | Imaging Q | Aesthetic Q |
|---|---|---|---|---|
| Top-1 | 11.1% | 0.1754 | 0.6928 | 0.5546 |
| Top-2 | 30.1% | 0.1466 | 0.7251 | 0.4871 |
| $\gamma$ | **Gen. Rate** | **Consistency** | **Imaging Q** | **Aesthetic Q** |
| 8 | 28.4% | 0.1943 | 0.6942 | 0.5475 |
| 10 | 11.1% | 0.1754 | 0.6928 | 0.5546 |
| 12 | 3.2% | 0.0933 | 0.6843 | 0.4987 |

Eraser on several identities, such as Biden and Elizabeth, while preserving a simpler and more deployment-friendly inference pipeline. These results suggest that CLEAR offers a more favorable trade-off between erasure effectiveness, inference efficiency, and practical deployability.

### 4.5. Results on Artist Style Concepts

We evaluate CLEAR on five representative artistic style erasure. Following ESD (Gandikota et al., 2023), we adopt

a CLIP-based evaluation framework (Lu et al., 2024) and introduce a Trade-off Score ($H_a$), measuring the net margin between content preservation ($VCLIP_s$) and style suppression ($VCLIP_e$). Higher $H_a$ indicates better style-content separation. CLEAR achieves the highest $H_a$ scores across all artists, demonstrating superior style-content disentanglement. For Rembrandt, CLEAR attains $H_a = 0.1308$, nearly doubling SAFREE (0.0731) and significantly surpassing T2VUnlearning (0.0906). This confirms CLEAR's precision in identifying and surgically removing style while preserving semantic content. Visual results are shown in Fig. 4, with more results provided in the Appendix I.

### 4.6. Ablation Study

We conduct ablation studies to systematically examine three aspects of CLEAR: (i) whether the proposed depth search can efficiently identify concept-aligned intervention layers, (ii) whether the separability-aware objective is necessary for avoiding reconstruction-driven but semantically entangled layers, and (iii) whether the resulting SAE-based intervention design remains effective under different inference configurations and evasive prompts.

**Efficiency and Topological Precision of Layer Search.** We compare CLEAR against a coarse Manual baseline that probes layers at stride-4 intervals, as well as brute-force layer-wise search. As shown in Table 5, CLEAR identifies intervention layers in a single training run (1×), whereas brute-force search requires 24× cost and the Manual baseline still incurs 5× cost. Despite this substantially lower search cost, CLEAR achieves comparable or better erasure–quality trade-offs while providing finer depth localization than the Manual baseline.

*Table 7.* **Layer-wise Ablation on Wan2.2-5B for Nudity Erasure.**

| Selected Layer | Origin | 2 | 4 | 6 | 8 | 10 |
|---|---|---|---|---|---|---|
| Gen. Rate | 67.3 % | 6.2 % | 5.1 % | 1.9 % | 11.2 % | 2.6 % |
| Consistency | 23.12 | 8.12 | 7.12 | 10.22 | 9.61 | 4.97 |
| **Selected Layer** | **12** | **14** | **16** | **18** | **20** | **22** |
| Gen. Rate | 4.6 % | 1.5 % | 4.2 % | 32.9 % | 13.4 % | 28.0 % |
| Consistency | 10.97 | 7.38 | 9.96 | 19.31 | 16.45 | 18.48 |

*Table 8.* **Ablation on SAE with Wan2.2-5B on Nudity.**

| Erasure Method | Gen. Rate | Consistency | Imaging Q | Aesthetic Q | Motion S |
|---|---|---|---|---|---|
| Steering Vector | 24.5% | 0.1986 | 0.6946 | 0.5240 | 0.9921 |
| CLEAR | 11.1% | 0.1754 | 0.6928 | 0.5546 | 0.9951 |

*Table 9.* **Nudity Erasure on Ring-A-Bell.**

| Erasure Method | Gen. Rate | Consistency | Imaging Q | Aesthetic Q | Motion S |
|---|---|---|---|---|---|
| Wan2.2 | 69.6% | 0.2216 | 0.6876 | 0.5541 | 0.9922 |
| CLEAR | 25.6% | 0.1227 | 0.7327 | 0.5373 | 0.9950 |
| CogX-2B | 31.8% | 0.2350 | 0.5070 | 0.4675 | 0.9937 |
| CLEAR | 22.2% | 0.1734 | 0.4633 | 0.4153 | 0.9857 |

Importantly, the selected layers exhibit concept-dependent semantic structure. For Parachute, CLEAR selects Block 6, which is close to the Manual optimum at Block 8 but obtained without repeated layer-wise trials. For Nudity, the Manual baseline attains stronger suppression but at the cost of severe visual degradation, whereas CLEAR selects a nearby deeper layer that better balances erasure and fidelity. For Springer Dog, CLEAR favors a shallow layer and achieves effective erasure with higher aesthetic quality, suggesting that texture-level representations can help suppress fine-grained object cues without disrupting deeper semantic structure.

**Impact of Separability-Aware Optimization.** We evaluate the role of separability-aware optimization by comparing CLEAR with a variant trained using reconstruction loss only (w/o $\mathcal{L}_{con}$). As shown in Table 5, removing $\mathcal{L}_{con}$ consistently biases the search toward intermediate layers (Blocks 13–15). Although these layers provide strong reconstruction fidelity, they do not exhibit sufficient concept–non-target separability. Consequently, the resulting interventions fail to suppress target concepts effectively, leading to substantially higher Generative Rates (e.g., 27.0% for Parachute and 14.0% for Springer Dog). These results indicate that reconstruction quality alone is not a reliable criterion for locating effective intervention depths. In contrast, $\mathcal{L}_{con}$ explicitly prioritizes concept–non-target separability, steering the search toward layers that support more precise and semantically localized suppression.

**Single-Layer versus Multi-Layer Intervention.** To examine whether multi-layer intervention improves erasure performance, we compare the Top-1 and Top-2 layer selections identified by CLEAR. As shown in Table 6, intervening only at the Top-1 layer achieves substantially lower Generative Rate than using both Top-1 and Top-2 layers simultaneously (11.1% vs. 30.1%). The single-layer intervention also better preserves overall consistency and aesthetic quality. These results suggest that multi-layer intervention introduces feature interference rather than complementary suppression, supporting the use of a single concept-aligned intervention layer during inference.

**Effect of Intervention Strength.** We further analyze the scaling factor $\gamma$, which controls sparse feature subtraction strength. As shown in Table 6, increasing $\gamma$ generally improves target suppression but reduces consistency and aesthetic quality. Among the tested settings, $\gamma = 10$ provides the best trade-off, reducing the Generative Rate to 11.1% while maintaining strong visual fidelity. This indicates that CLEAR provides controllable intervention strength that can be adjusted according to deployment requirements.

**SAE-based Intervention versus Dense Steering.** To assess the need for sparse decomposition, we compare CLEAR with a steering-vector baseline built from the mean activation difference between positive and negative prompts. As shown in Table 8, this baseline yields weaker suppression (24.5% vs. 11.1%) and lower aesthetic quality (0.5240 vs. 0.5546). It suggests that dense activation differences remain entangled with shared semantic factors, making precise removal difficult. By contrast, the SAE provides a sparse, semantically localized basis for selective intervention.

**Generalization to Evasive Prompts.** Finally, we evaluate CLEAR on the Ring-A-Bell (Tsai et al., 2024) benchmark, which uses paraphrased prompts designed to evoke target concepts without explicitly naming them. As shown in Table 9, CLEAR consistently reduces the Generative Rate on both backbones (Wan2.2-5B: 69.6% → 25.6%; CogVideoX-2B: 31.8% → 22.2%). Although suppression is naturally weaker than on explicit prompts, the reduction demonstrates that CLEAR intervenes on concept-level representations rather than merely suppressing surface-level lexical triggers.

## 5. Conclusion

We presented CLEAR, a framework for concept erasure in text-to-video diffusion models. By jointly optimizing *where* to intervene and *how* to erase, CLEAR combines differentiable depth selection with sparse, concept-aligned feature manipulation. Experiments on Wan2.2-5B and CogVideoX-2B demonstrate effective removal of sensitive and identity-specific concepts while preserving, and sometimes improving, visual fidelity. With an order-of-magnitude reduction in search cost, CLEAR offers a scalable solution for safe deployment of large-scale video generative models.

## Impact Statement

This work studies concept erasure for text-to-video diffusion models, with the goal of improving the safety and controllability of generative video systems. By enabling targeted suppression of sensitive, identity-specific, or copyrighted concepts without retraining the full model, CLEAR may help reduce harmful or unauthorized content generation in deployed systems. At the same time, concept erasure techniques could be misused to suppress benign or socially important content, or to alter model behavior in non-transparent ways. We therefore emphasize that such methods should be deployed with clear policy definitions, auditing procedures, and transparency about which concepts are being restricted.

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

## A. Algorithm Pseudocode

We provide the detailed training procedure of CLEAR in Algorithm 1. The optimization follows a bilevel strategy: in the inner loop, we update the SAE weights to reconstruct the residual features; in the outer loop, we update the architecture parameters $\alpha$ to minimize the concept separability loss $\mathcal{L}_{con}$.

---

**Algorithm 1** Concept Erasure via Differentiable Architecture Search (CLEAR)

---

**Require:** Pre-trained T2V model $\mathcal{M}$, Target Concept $C$, Positive Prompts $P_{pos}$, Negative Prompts $P_{neg}$
**Require:** Search Space $\mathcal{S}$ (Candidate Layers), SAE Dictionary $\mathcal{D}$
1: **Initialize:** Architecture parameters $\alpha$ (uniform distribution), SAE weights $\theta$
2: **Hyperparameters:** Temperature $\tau_{max} = 1.0$, $\tau_{min} = 0.1$, Iterations $T_{max}$
3: **while** $t < T_{max}$ **do**
4:     Anneal temperature: $\tau \leftarrow \tau_{max} - (\tau_{max} - \tau_{min}) \cdot t/T_{max}$
5:     *// Step 1: Sample Architecture (Forward Pass)*
6:     Sample layer index $k \sim$ Gumbel-Softmax$(\alpha, \tau)$
7:     Extract activations $h_{k-pos}$ and $h_{k-neg}$ from layer $k$ given $P_{pos}$ and $P_{neg}$
8:     *// Step 2: Inner Loop (Train SAE)*
9:     Compute mixed input: $\hat{h}_{pos} = \sum_{i=0}^{L} h_{i-pos}$
10:    Decode features: $\hat{h}_{pos} = \text{SAE}(h_{pos}; \theta)$
11:    Compute Reconstruction Loss: $\mathcal{L}_{rec} = \|h_{pos} - \hat{h}_{pos}\|_2^2 + \lambda\|f\|_1$
12:    Update $\theta \leftarrow \theta - \eta_\theta \nabla_\theta \mathcal{L}_{rec}$
13:    *// Step 3: Outer Loop (Architecture Search)*
14:    Compute Soft-Threshold Contrastive Loss $\mathcal{L}_{con}$ based on feature specificity
15:    Update $\alpha \leftarrow \alpha - \eta_\alpha \nabla_\alpha \mathcal{L}_{CLEAR}$
16:    $t \leftarrow t + 1$
17: **end while**
18: **Output:** Optimal intervention layer $k^* = \arg\max(\alpha)$

---

## B. Detailed Implementation Setup

All experiments are conducted on NVIDIA H100 GPU. The specific hyperparameters for the Sparse Autoencoder (SAE) training and the CLEAR search process are detailed in Table 10. Our unified SAE is configured with a hidden dimension of $131,072$, utilizing a standard $\ell_1$ penalty to induce sparsity. To ensure robust generalization across diverse semantic contexts, we construct a comprehensive training dataset using prompts synthesized by a Large Language Model (LLM). Each pair shares identical semantics, differing only by the target concept. For example, targeting "Van Gogh": Positive: "Classical figures with contemporary fragmented forms, with Van Gogh's starry night style." Negative: "Classical figures with contemporary fragmented forms, in Caravaggio's naturalistic style." No test prompts (from T2VUnlearning benchmark) appear in training/validation data. For the architecture search, the CLEAR process is optimized via Adam, with the Gumbel-Softmax temperature $\tau$ annealed linearly from $1.0$ to $0.1$ to stabilize convergence. We tailor the optimization schedule to the complexity of the target: the sensitive *nudity* concept is trained for $3,750$ iterations to ensure thorough erasure, while all other concepts are trained for $2,500$ iterations. Crucially, the entire pipeline—encompassing both topological search and SAE parameter learning—is highly efficient, completing in approximately two hours on a single H100 GPU.

*Table 10.* Hyperparameter settings for CLEAR.

| Hyperparameter | Value |
|---|---|
| SAE Hidden Dimension | 131,072 |
| Sparsity Coefficient ($\lambda$) | $1e^{-4}$ |
| Optimizer | Adam |
| Learning Rate ($\alpha$, Architecture) | $3e^{-2}$ |
| Learning Rate ($\theta$, SAE Weights) | $1e^{-3}$ |
| Batch Size | 16 |
| Total Iterations (Nudity) | 3,750 |
| Total Iterations (Objects/Celebrities) | 2,500 |
| Gumbel Temperature Schedule | Linear Decay ($1.0 \rightarrow 0.1$) |
| Prompt Source | Synthesized by Llama-3-8B |

## C. Architecture and Training Pipeline

The SAE and NAS modules are trained independently per concept. This decoupled paradigm is essential because distinct concepts exhibit unique geometric entanglement patterns and optimal depth preferences, necessitating concept-specific disentanglement. During the search phase for a given concept, a single shared SAE is applied across all candidate layers.

*Table 11.* **Linear Probe Error Rate Across T5 Blocks with Wan2.2-5B on Nudity.**

| Block | 0 | 1 | 2 | 3 | 4 | 5 | 6 | 7 | 8 | 9 | 10 | 11 |
|---|---|---|---|---|---|---|---|---|---|---|---|---|
| Error (%) | 4.5 | 4.2 | 4.5 | 4.3 | 4.3 | 4.2 | 4.4 | 4.1 | 4.4 | 3.8 | 3.3 | 2.4 |
| **Block** | **12** | **13** | **14** | **15** | **16** | **17** | **18** | **19** | **20** | **21** | **22** | **23** |
| Error (%) | 1.6 | 1.3 | 1.0 | 0.9 | 0.6 | 0.6 | 0.5 | 0.5 | 0.4 | **0.4** | 0.5 | 0.5 |

*Table 12.* **Comprehensive breakdown of preservation metrics across all object categories.** We detail the *Overall Consistency*, *Imaging Quality*, and *Aesthetic Quality* for each of the 10 object concepts on both Wan2.2-5B and CogVideoX-2B.

*(a)* Wan2.2-5B: Overall Consistency

| Erasure Method | Cassette player | Chain saw | Church | Springer dog | French horn | Garbage Truck | Gas pump | Golf ball | Parachute | Tench |
|---|---|---|---|---|---|---|---|---|---|---|
| Origin | 0.2792 | 0.2415 | 0.2622 | 0.2414 | 0.2656 | 0.2447 | 0.2832 | 0.2567 | 0.2364 | 0.1792 |
| NegPrompt | **0.2914** | **0.2515** | **0.2655** | **0.2507** | **0.2569** | **0.2551** | **0.2883** | **0.2556** | **0.2394** | **0.2013** |
| SAFREE | 0.2443 | 0.1968 | 0.2324 | 0.2201 | 0.2301 | 0.2313 | 0.2457 | 0.2261 | 0.2355 | 0.1492 |
| T2VUnlearning | 0.1562 | 0.2446 | 0.2478 | 0.2195 | 0.2348 | 0.1110 | 0.2880 | 0.1232 | 0.1097 | 0.1700 |
| CLEAR | 0.1908 | 0.2092 | 0.2266 | 0.2027 | 0.1773 | 0.1880 | 0.2160 | 0.1472 | 0.2073 | 0.1307 |

*(b)* Wan2.2-5B: Imaging Quality

| Erasure Method | Cassette player | Chain saw | Church | Springer dog | French horn | Garbage Truck | Gas pump | Golf ball | Parachute | Tench |
|---|---|---|---|---|---|---|---|---|---|---|
| Origin | 0.7087 | 0.7385 | 0.7595 | 0.6556 | 0.6631 | 0.7435 | 0.7073 | 0.5692 | 0.6776 | 0.6866 |
| NegPrompt | 0.6532 | 0.6942 | **0.7372** | 0.6172 | 0.6603 | 0.7030 | 0.7078 | 0.5390 | 0.6957 | 0.6361 |
| SAFREE | 0.6269 | 0.6225 | 0.6922 | 0.5935 | 0.6246 | 0.5956 | 0.6424 | 0.6142 | 0.6784 | 0.6138 |
| T2VUnlearning | 0.6971 | 0.6929 | 0.7041 | 0.6314 | 0.6453 | 0.6027 | 0.7274 | 0.5991 | 0.7172 | 0.6348 |
| CLEAR | **0.7096** | **0.7328** | 0.7289 | **0.6421** | **0.6761** | **0.7358** | **0.7306** | **0.6672** | **0.7255** | **0.6759** |

*(c)* Wan2.2-5B: Aesthetic Quality

| Erasure Method | Cassette player | Chain saw | Church | Springer dog | French horn | Garbage Truck | Gas pump | Golf ball | Parachute | Tench |
|---|---|---|---|---|---|---|---|---|---|---|
| Origin | 0.6337 | 0.5703 | 0.6660 | 0.6056 | 0.5361 | 0.5863 | 0.6472 | 0.5386 | 0.5923 | 0.6051 |
| NegPrompt | **0.6261** | 0.5468 | **0.6564** | **0.6106** | **0.5271** | 0.5671 | **0.6547** | 0.5089 | 0.5953 | **0.5888** |
| SAFREE | 0.5709 | 0.4967 | 0.6219 | 0.5839 | 0.4986 | 0.5633 | 0.6483 | 0.5052 | 0.5715 | 0.5267 |
| T2VUnlearning | 0.5125 | **0.5602** | 0.6215 | 0.5643 | 0.4809 | 0.4856 | 0.6473 | 0.3741 | 0.4361 | 0.5147 |
| CLEAR | 0.5525 | 0.5177 | 0.6503 | 0.6048 | 0.5124 | **0.5795** | 0.6388 | **0.5198** | **0.6010** | 0.5809 |

*(d)* CogVideoX-2B: Overall Consistency

| Erasure Method | Cassette player | Chain saw | Church | Springer dog | French horn | Garbage Truck | Gas pump | Golf ball | Parachute | Tench |
|---|---|---|---|---|---|---|---|---|---|---|
| Origin | 0.2604 | 0.2604 | 0.2582 | 0.1588 | 0.2868 | 0.2320 | 0.2671 | 0.2630 | 0.2144 | 0.2461 |
| NegPrompt | **0.2449** | **0.2424** | **0.2499** | **0.1577** | **0.2612** | **0.2170** | **0.2663** | **0.2667** | **0.2198** | **0.2459** |
| SAFREE | 0.2129 | 0.1867 | 0.1936 | 0.1047 | 0.2277 | 0.1938 | 0.1980 | 0.2161 | 0.2085 | 0.1821 |
| T2VUnlearning | 0.2169 | 0.0802 | 0.2321 | 0.1572 | 0.2018 | 0.0643 | 0.1492 | 0.2322 | 0.1787 | 0.2381 |
| CLEAR | 0.1574 | 0.1638 | 0.2485 | 0.1184 | 0.1635 | 0.1393 | 0.1927 | 0.1948 | 0.2141 | 0.1548 |

*(e)* CogVideoX-2B: Imaging Quality

| Erasure Method | Cassette player | Chain saw | Church | Springer dog | French horn | Garbage Truck | Gas pump | Golf ball | Parachute | Tench |
|---|---|---|---|---|---|---|---|---|---|---|
| Origin | 0.4976 | 0.5346 | 0.6077 | 0.5632 | 0.5327 | 0.6216 | 0.6007 | 0.4942 | 0.5758 | 0.4769 |
| NegPrompt | **0.4902** | **0.5138** | 0.5336 | **0.5429** | **0.4503** | **0.5103** | **0.5877** | 0.4575 | 0.5321 | **0.4879** |
| SAFREE | 0.4398 | 0.4260 | 0.4543 | 0.5053 | 0.4068 | 0.4167 | 0.4041 | **0.5587** | 0.5121 | 0.3986 |
| T2VUnlearning | 0.4843 | 0.3790 | 0.4718 | 0.3708 | 0.2896 | 0.2296 | 0.5407 | 0.3404 | 0.3812 | 0.4192 |
| CLEAR | 0.3728 | 0.4148 | **0.5875** | 0.4784 | 0.3300 | 0.4678 | 0.5388 | 0.5394 | **0.5708** | 0.3786 |

*(f)* CogVideoX-2B: Aesthetic Quality

| Erasure Method | Cassette player | Chain saw | Church | Springer dog | French horn | Garbage Truck | Gas pump | Golf ball | Parachute | Tench |
|---|---|---|---|---|---|---|---|---|---|---|
| Origin | 0.5123 | 0.4790 | 0.5917 | 0.5380 | 0.4126 | 0.5204 | 0.5996 | 0.4760 | 0.4758 | 0.4948 |
| NegPrompt | **0.4830** | **0.4768** | **0.5666** | **0.5259** | 0.3717 | 0.4636 | **0.5745** | **0.4466** | **0.4793** | **0.4900** |
| SAFREE | 0.4591 | 0.4030 | 0.4775 | 0.4812 | 0.3657 | 0.4215 | 0.5066 | 0.4257 | 0.4527 | 0.4071 |
| T2VUnlearning | 0.4784 | 0.3735 | 0.5090 | 0.4456 | 0.3376 | 0.2618 | 0.4494 | 0.3880 | 0.3883 | 0.4498 |
| CLEAR | 0.4221 | 0.3700 | 0.5573 | 0.4937 | **0.3758** | **0.4641** | 0.4651 | 0.4375 | 0.4756 | 0.4229 |

*Table 13.* **Motion Smoothness metrics** across all object categories for each of the 10 object concepts on both Wan2.2-5B and CogVideoX-2B.

*(a)* Wan2.2-5B: Motion Smoothness

| Erasure Method | Cassette player | Chain saw | Church | Springer dog | French horn | Garbage Truck | Gas pump | Golf ball | Parachute | Tench |
|---|---|---|---|---|---|---|---|---|---|---|
| Origin | 0.9936 | 0.9848 | 0.9954 | 0.9724 | 0.9872 | 0.9854 | 0.9955 | 0.9952 | 0.9934 | 0.9835 |
| NegPrompt | **0.9927** | **0.9801** | **0.9948** | **0.9551** | **0.9839** | **0.9807** | **0.9956** | **0.9955** | **0.9936** | **0.9821** |
| SAFREE | 0.9935 | 0.9895 | 0.9924 | 0.9717 | 0.9848 | 0.9891 | 0.9940 | 0.9941 | 0.9891 | 0.9895 |
| T2VUnlearning | 0.9884 | 0.9762 | 0.9933 | 0.9721 | 0.9832 | 0.9903 | 0.9960 | 0.9772 | 0.9960 | 0.9748 |
| CLEAR | 0.9961 | 0.9859 | 0.9953 | 0.9812 | 0.9893 | 0.9804 | 0.9953 | 0.9897 | 0.9925 | 0.9808 |

*(b)* CogVideoX-2B: Motion Smoothness

| Erasure Method | Cassette player | Chain saw | Church | Springer dog | French horn | Garbage Truck | Gas pump | Golf ball | Parachute | Tench |
|---|---|---|---|---|---|---|---|---|---|---|
| Origin | 0.9903 | 0.9723 | 0.9926 | 0.9590 | 0.9491 | 0.9804 | 0.9895 | 0.9914 | 0.9804 | 0.9694 |
| NegPrompt | **0.9905** | **0.9807** | **0.9914** | **0.9601** | **0.9480** | **0.9715** | **0.9904** | **0.9906** | **0.9737** | **0.9640** |
| SAFREE | 0.9871 | 0.9802 | 0.9859 | 0.9657 | 0.9595 | 0.9714 | 0.9874 | 0.9890 | 0.9710 | 0.9658 |
| T2VUnlearning | 0.9916 | 0.9940 | 0.9891 | 0.9198 | 0.9593 | 0.8494 | 0.9745 | 0.9858 | 0.9543 | 0.9267 |
| CLEAR | 0.9900 | 0.9733 | 0.9926 | 0.9601 | 0.9697 | 0.9750 | 0.9848 | 0.9906 | 0.9805 | 0.9615 |

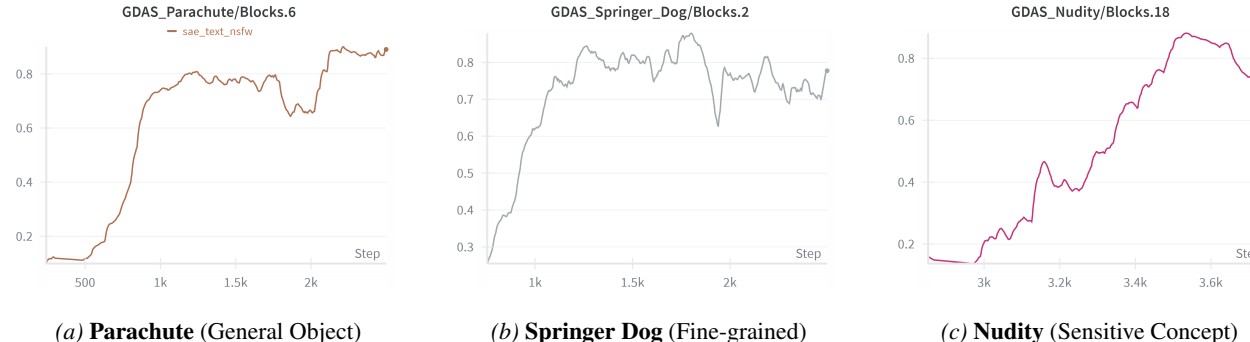

*(a)* **Parachute** (General Object)          *(b)* **Springer Dog** (Fine-grained)          *(c)* **Nudity** (Sensitive Concept)

*Figure 5.* **Evolution of layer selection probabilities during the CLEAR search process.** The x-axis represents the training iterations (0 to 2500/3750), and the y-axis represents the probability of the dominant layer. As the Gumbel-Softmax temperature anneals, the probability distribution sharpens from a uniform initialization to a deterministic selection, pinpointing distinct topological depths for different concepts.

This parameter sharing is natively supported by the constant residual stream dimension ($d_{\text{model}}$) across all text encoder layers, eliminating the need for intermediate projections. As the NAS module routes representations from various candidate layers through this shared SAE, the Gumbel-Softmax temperature gradually anneals. Consequently, the SAE progressively co-adapts and specializes to the latent distribution of the final converged layer.

## D. Visualization of Topological Dependency

A core finding of our work is that different concepts are localized at different depths within the T2V model. Figure 5 visualizes the probability evolution of the architecture parameters ($\alpha$) during the search with Wan2.2-5B. We observe distinct convergence trajectories: general objects like *Parachute* localize in shallower layers (e.g., Block 6), while complex sensitive concepts like *Nudity* require deeper interventions (e.g., Block 18). This empirical evidence validates our hypothesis that concept erasure requires depth-aware interventions governed by semantic complexity.

## E. Layer-wise Linear Probe Analysis of Conceptual Divisibility

In the main text, we visualized the distribution of positive and negative samples of explicit content prior to SAE processing; the results, shown in Figure 2, demonstrate a high degree of conceptual indistinguishability. To further investigate conceptual separability across different layers, we trained linear probes (logistic regression) on each T5 layer to distinguish between positive and negative conceptual prompts. The error rates for explicit content on the Wan2.2-5B dataset are shown in

Table. 11. The results validate concept-layer alignment: while all layers achieve above-chance accuracy, the error rate drops by an order of magnitude from shallow layers (Blocks 0-8, error 4.5%) to deeper layers (Block 21, error 0.4%), indicating substantially stronger linear separability at depth. CLEAR selects Block 18 (error 0.5%), near the minimum-error depth. This offset reflects CLEAR's joint objective ($\mathcal{L}_{con} + \mathcal{L}_{SAE}$), which balances separability with reconstruction quality rather than optimizing separability alone. The sharp transition in error rate from Blocks 8-12 is consistent with the depth-dependent separability described in our theoretical account.

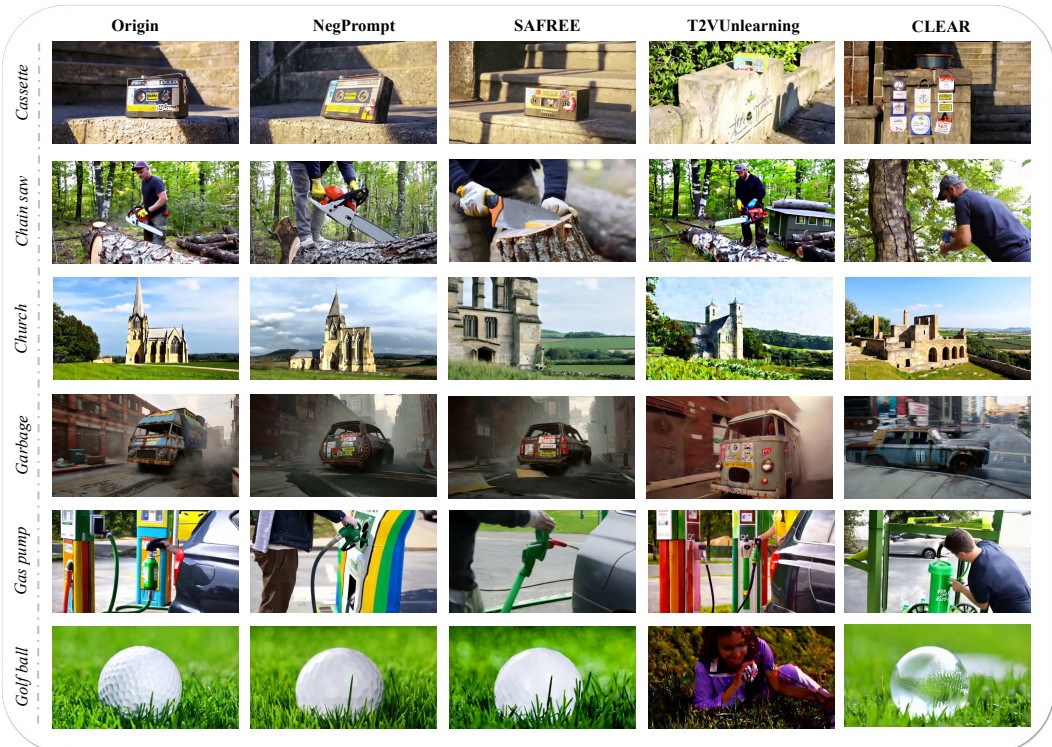

*Figure 6.* **Qualitative comparison of object concepts on Wan2.2-5B.** Comparing Origin, NegPrompt, T2VUnlearning, and CLEAR.

## F. Additional Results on Objects Concepts

Due to space constraints in the main text, we presented averaged metrics of overall consistency, imaging quality and aesthetic quality on objects concepts. Here, we provide the detailed breakdown of erasure performance across all 10 individual object categories for the Wan2.2-5B and CogX-2B model in Table 12. CLEAR preserves a superior model overall utility. The parameter-update baseline (T2VUnlearning) often sacrifices Imaging Quality to achieve erasure, CLEAR preserves high fidelity across all categories. This confirms that our adaptive depth search successfully isolates concept-specific features without damaging the model's general backbone.

Figure 6 further corroborates these metrics. While inference-time defenses (NegPrompt) often exhibit "ghosting" artifacts (high consistency but poor erasure), and global unlearning methods induce contrast degradation (low quality), CLEAR achieves *clean semantic replacement*. For instance, it seamlessly transforms a target object into a semantically plausible alternative (e.g., bird or backpack) while preserving the high-frequency details and lighting of the scene.

## G. Additional Results on Nudity Concepts

We provide extensive visual comparisons in Figure 7. Since nudity is inappropriate to display directly, we have blurred the output from the various methods.

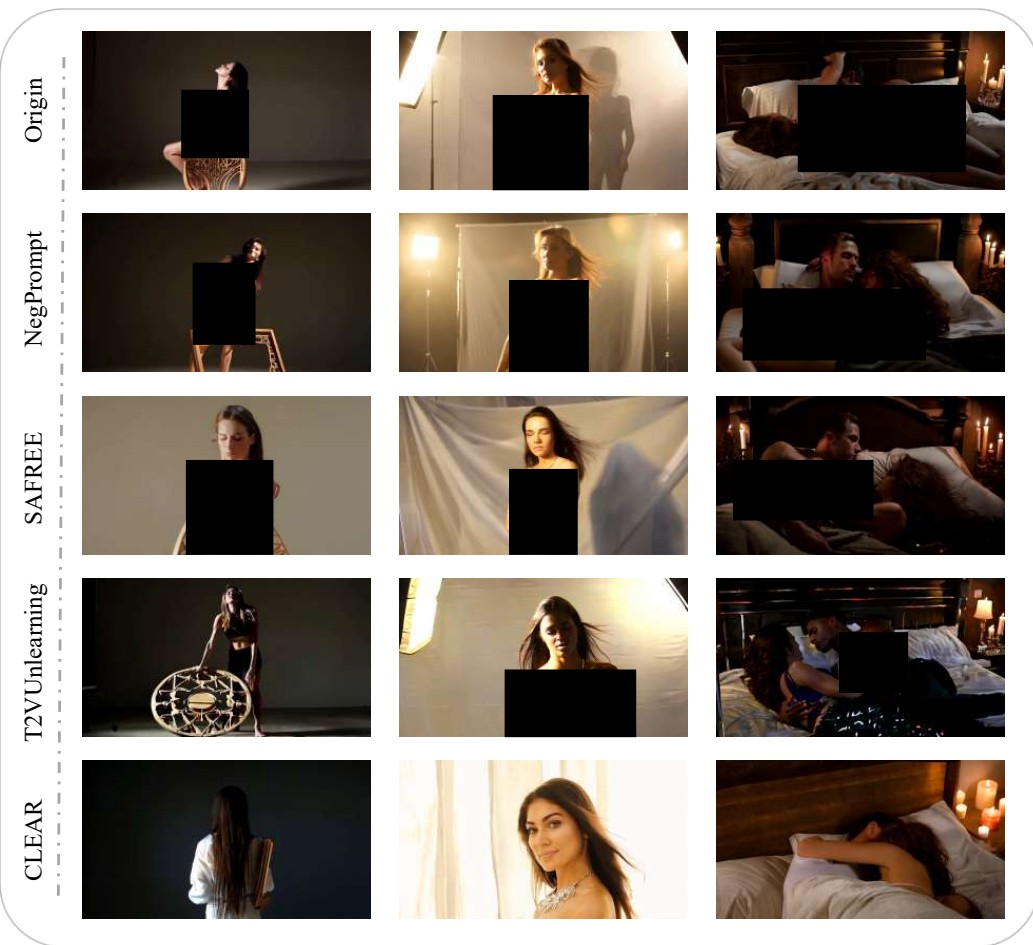

*Figure 7.* **Qualitative comparison of nudity concepts on Wan2.2-5B.** Comparing Origin, NegPrompt, T2VUnlearning, and CLEAR.

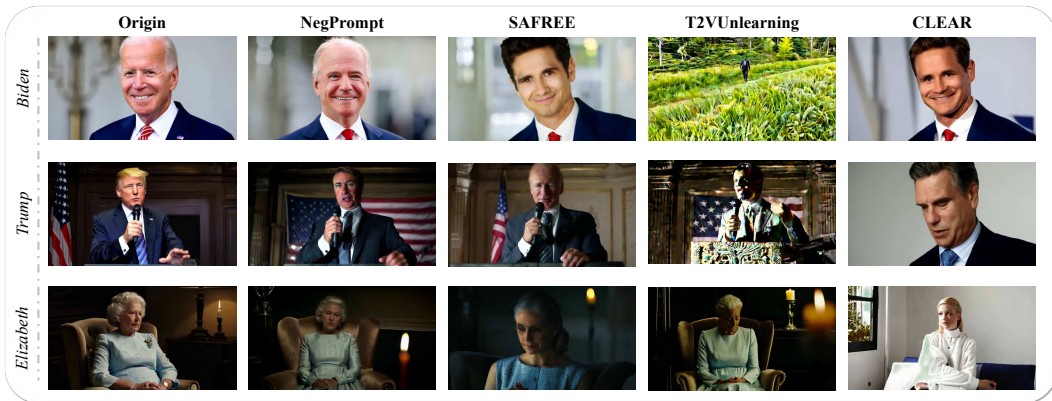

*Figure 8.* **Qualitative comparison of celebrity concepts on Wan2.2-5B.** Comparing Origin, NegPrompt, T2VUnlearning, and CLEAR.

## H. Additional Results on Celebrity Concepts

We provide extensive visual comparisons in Figure 8. Compared with existing baselines, CLEAR more effectively suppresses identity-specific facial characteristics while preserving scene composition, pose, and overall visual realism. In contrast, inference-time baselines often retain recognizable identity cues, whereas parameter-update methods may introduce noticeable visual distortion or semantic drift.

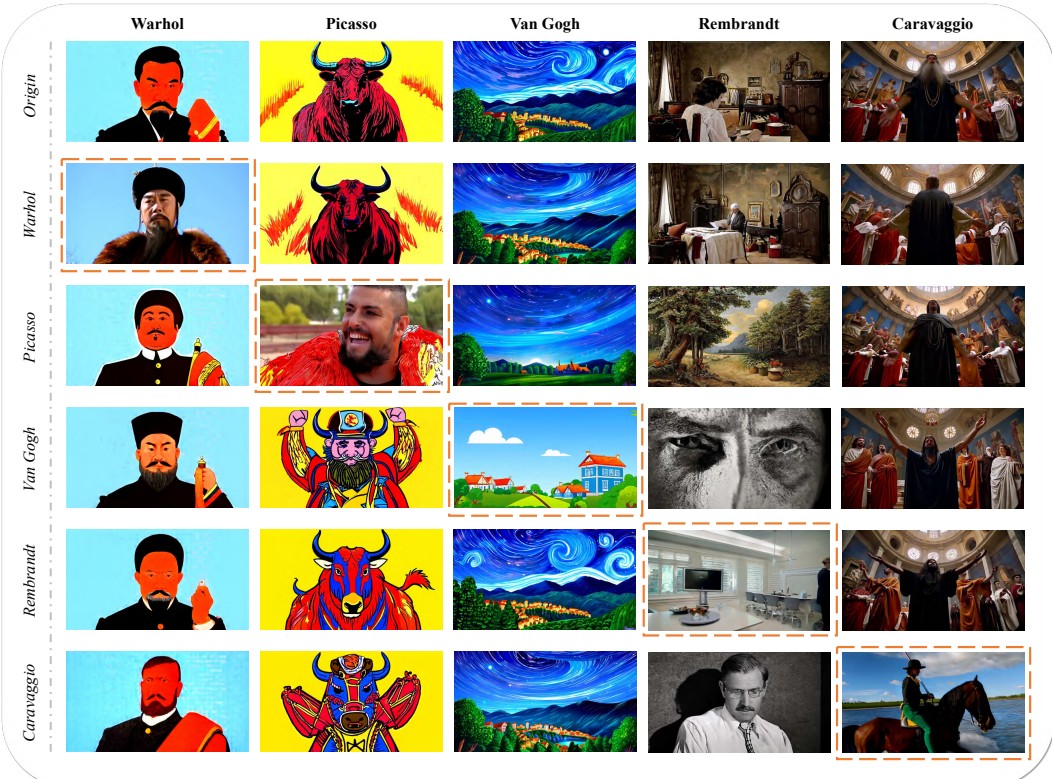

*Figure 9.* **Qualitative comparison of artist styles concepts on Wan2.2-5B under CLEAR.** We present a generation matrix to evaluate the precision of CLEAR in disentangling different artistic styles. Columns correspond to prompts conditioning on five distinct artists (Andy Warhol, Picasso, Van Gogh, Rembrandt, Caravaggio). Rows indicate the specific erasure intervention applied (e.g., the second row "Warhol" shows results when the model is optimized to erase Andy Warhol). The orange dashed boxes along the diagonal highlight the successful removal of the target style, where the images revert to a generic or photorealistic appearance (e.g., the pop-art style of Warhol is removed).

*Table 14.* **Quantitative evaluation of artist style erasure on CogX-2B.** We report the Video-CLIP score for *Style Erasure* ($VCLIP_e$, lower is better) and *Semantic Preservation* ($VCLIP_s$, higher is better). $H_a$ denotes the *Trade-off Score* ($H_a = VCLIP_s - VCLIP_e$), which quantifies the net effectiveness of the intervention.

| Erasure Method | Pablo Picasso | | | Van Gogh | | | Rembrandt | | | Andy Warhol | | | Caravaggio | | |
|---|---|---|---|---|---|---|---|---|---|---|---|---|---|---|---|
| | $VCLIP_e \downarrow$ | $ViCLIP_s \uparrow$ | $H_a \uparrow$ | $VCLIP_e \downarrow$ | $VCLIP_s \uparrow$ | $H_a \uparrow$ | $VCLIP_e \downarrow$ | $VCLIP_s \uparrow$ | $H_a \uparrow$ | $VCLIP_e \downarrow$ | $VCLIP_s \uparrow$ | $H_a \uparrow$ | $VCLIP_e \downarrow$ | $VCLIP_s \uparrow$ | $H_a \uparrow$ |
| Origin | 0.2487 | 0.2341 | / | 0.2709 | 0.2285 | / | 0.2190 | 0.2415 | / | 0.1992 | 0.2465 | / | 0.2472 | 0.2345 | / |
| NegPrompt | 0.2364 | **0.2266** | -0.0368 | 0.2503 | **0.2180** | -0.0323 | 0.2114 | **0.2353** | 0.0239 | 0.1797 | **0.2508** | 0.0711 | 0.2183 | 0.2130 | -0.0053 |
| SAFREE | 0.1784 | 0.1979 | 0.0195 | 0.1902 | 0.1919 | 0.0017 | 0.1323 | 0.2065 | 0.0742 | 0.1219 | 0.2212 | 0.0993 | 0.1729 | 0.1882 | **0.0153** |
| T2VUnlearning | **0.1629** | 0.1832 | **0.0203** | 0.1577 | 0.1924 | **0.0347** | 0.0546 | 0.1597 | 0.1051 | **0.0800** | 0.2328 | **0.1528** | 0.2017 | **0.2168** | 0.0151 |
| CLEAR | 0.1915 | 0.1737 | -0.0178 | **0.1555** | 0.1592 | 0.0037 | 0.0639 | 0.2176 | **0.1537** | 0.1449 | 0.2338 | 0.0889 | **0.1601** | 0.1606 | 0.0005 |

# I. Additional Results on Artist Style Concepts

We further evaluate the generalization capability of CLEAR on abstract artist style erasure using the CogVideoX-2B model, with quantitative results reported in Table 14 and visual comparisons on Wan2.2-5B provided in Figure 9.

# J. Multi-concept Erasure Experiments

Table 15 demonstrates CLEAR's robustness in multi-concept settings. In intra-category tasks (e.g., erasing Van Gogh and Picasso simultaneously), baselines like T2VUnlearning suffer from severe catastrophic forgetting, indiscriminately degrading un-targeted related concepts (e.g., Rembrandt's VCLIP score plunges to 0.0283). Conversely, CLEAR preserves high fidelity for retained concepts (0.1315 for Rembrandt) by leveraging DAS for precise layer isolation, which effectively decouples parameter updates and prevents subspace entanglement. Furthermore, in cross-category scenarios, CLEAR can

*Table 15.* **Multi-Concept Erasure.**

| *SameCategory (Van Gogh & Picasso)* | | | | | |
|---|---|---|---|---|---|
| **Erasure Method** | **Van Gogh** | **Picasso** | **Andy Warhol** | **Caravaggio** | **Rembrandt** |
| Origin | 0.2388 | 0.1804 | 0.1450 | 0.1945 | 0.1566 |
| T2VUnlearning | 0.1719 | 0.1624 | 0.0938 | 0.1137 | 0.0283 |
| CLEAR | 0.1472 | 0.1442 | 0.1268 | 0.1924 | 0.1315 |
| *CrossCategory MultiConcept Erasure* | | | | | |
| **Erasure Method** | **Van Gogh (VCLIP$_e$)** | **Parachute (Gen.Rate** %**)** | | | |
| Origin | 0.2388 | 89.6 | | | |
| CLEAR | 0.1268 | 17.4 | | | |

*Table 16.* **Migration to T2I and CLIP-based T2V model.**

| **Erasure Model** | **Generative Rate** | **Imaging Quality** | **Aesthetic Quality** |
|---|---|---|---|
| Text2Video-Zero | 70.4% | 65.95 | 0.5492 |
| CLEAR | 12.9% | 72.16 | 0.5121 |
| Flux 1.0-dev | 74.0 % | | |
| CLEAR | 18.0 % | | |

jointly suppress heterogeneous concepts, including an artistic style (Van Gogh: 0.2388 → 0.1268) and a specific object (Parachute: 89.6% → 17.4%). These results suggest that CLEAR can support multi-concept intervention across different semantic categories while maintaining reasonable preservation of non-target generation quality.

## K. Transferability to Text-to-Image Models

Since CLEAR operates on text-encoder representations, its intervention mechanism is not inherently tied to video generation. To examine its transferability, we extend CLEAR to FLUX.1-dev, a text-to-image diffusion model. As shown in Table 16, CLEAR reduces the nudity generation rate from 74.0% to 18.0%. This result suggests that text-encoder-level intervention may generalize beyond T2V models, although broader validation across architectures and concepts remains future work.

## L. Generalization Across Text Encoders

To examine whether CLEAR depends on a specific text encoder, we evaluate it on Text2Video-Zero (Khachatryan et al., 2023), which uses a CLIP-based text encoder. As shown in Table 16, CLEAR reduces the target generation rate from 70.4% to 12.9%. This result suggests that CLEAR is not limited to the T5-based text encoders used in our main experiments and may extend to alternative text-encoding architectures. Further evaluation across more text encoders and model families remains an important direction for future work.

## M. Extended Analysis on Multi-Layer Intervention.

To further investigate the efficacy of multi-layer intervention, we conducted additional experiments by varying the second layer's intervention strength independently, while fixing the Top-1 layer at $\gamma = 10$ (Table 17). Across all five configurations, the single-layer intervention (Top-1 only) consistently outperforms joint Top-1 & Top-2 interventions in generative rate (11.1% vs. 13.9%–30.1%), overall consistency (0.1754 vs. 0.1313–0.1466), and aesthetic quality (0.5546 vs. 0.4871–0.5173), while maintaining comparable imaging quality. This simultaneous degradation across primary metrics confirms that multi-layer intervention induces feature interference rather than complementary suppression. Furthermore, analyzing the routing weights learned during training reveals that the layer preference distribution naturally converges to a near-one-hot state ($> 90\%$ on a single layer, Appendix Fig. 5), assigning $< 5\%$ weight to the second layer. This intrinsic architectural convergence explicitly aligns with our empirical findings, theoretically justifying our single-layer deployment.

## N. Sensitivity to Training Data Volume

We study how the number of prompt pairs affects the erasure–preservation trade-off, using Parachute on Wan2.2-5B as a representative object concept. As shown in Table 18, different data scales lead to different balances between suppression strength and preservation quality. Using fewer prompt pairs can produce aggressive suppression, but it also lowers overall consistency, suggesting that limited semantic coverage may lead to less stable feature decomposition. Increasing the number of prompt pairs improves semantic coverage, but excessive or redundant prompts do not necessarily improve the trade-off

*Table 17.* **Comparing different $\gamma$ influences multi-layer erasure effectiveness.**

| Top-1 $\gamma$ | Top-2 $\gamma$ | Gen. Rate | Consistency | Imaging Q | Aesthetic Q |
|---|---|---|---|---|---|
| CLEAR ($\gamma$=10) | - | 11.1% | 0.1754 | 0.6928 | 0.5546 |
| 10 | 6 | 20.7% | 0.1424 | 0.7127 | 0.5173 |
| 10 | 8 | 22.5% | 0.1389 | 0.7174 | 0.5169 |
| 10 | 10 | 30.1% | 0.1466 | 0.7251 | 0.4871 |
| 10 | 12 | 16.8% | 0.1325 | 0.7136 | 0.5169 |
| 10 | 14 | 13.9% | 0.1313 | 0.7201 | 0.5063 |

*Table 18.* **Comparing different amount of train data influences erasure effectiveness.**

| Prompt Pairs | Gen. Rate | Consistency | Imaging Q | Aesthetic Q |
|---|---|---|---|---|
| 500 | 8.2% | 0.1406 | 0.6955 | 0.5591 |
| 1000 | 15.0% | 0.1214 | 0.7715 | 0.5678 |
| 2000 | 16.8% | 0.2159 | 0.7328 | 0.5902 |
| 3000 | 10.2% | 0.1469 | 0.7335 | 0.5479 |

*Table 19.* **Applying multiple intervention at the same layer (Block.18).**

| Objects | Consistency | Imaging Q | Aesthetic Q |
|---|---|---|---|
| Church | 0.2542 | 0.7492 | 0.6613 |
| French horn | 0.2331 | 0.6753 | 0.5051 |
| Garbage truck | 0.2227 | 0.7476 | 0.5969 |
| Gas pump | 0.2046 | 0.7216 | 0.6192 |
| Golf ball | 0.2287 | 0.5879 | 0.5066 |
| Parachute | 0.2186 | 0.7477 | 0.6002 |
| Springer dog | 0.2306 | 0.6953 | 0.6115 |
| Tench | 0.1548 | 0.6823 | 0.5796 |
| CLEAR Avg. | 0.2184 | 0.7009 | 0.5851 |
| Origin Avg. | 0.2462 | 0.6828 | 0.5972 |

and may introduce additional variability in the learned SAE features. Among the tested settings, the 2000-pair SAE training split achieves the best preservation metrics, with the highest overall consistency (0.2159) and aesthetic quality (0.5902), while still maintaining effective concept suppression (16.8%). These results suggest that CLEAR is reasonably robust to training data scale, but an intermediate amount of diverse prompt supervision provides the most favorable balance.

## O. Multiple Interventions within a Single Layer

To investigate whether applying multiple interventions at the same layer induces excessive latent perturbation and degrades overall generation quality, we simultaneously suppress two distinct concepts ("Cassette Player" and "Chain Saw") whose optimal SAE features both route to Block 18. Evaluating generation quality across un-targeted concepts (Table 19) reveals that same-layer multi-concept erasure avoids significant degradation. Imaging quality remains comparable to the original model (0.7009 vs. 0.6828), with only marginal reductions in text-image consistency (0.2184 vs. 0.2462) and aesthetic quality (0.5851 vs. 0.5972). We attribute this robustness to the sparse feature decomposition of SAEs: distinct concepts are encoded by largely non-overlapping dictionary elements. Consequently, combining multiple erasure vectors within the exact same layer minimizes cross-concept interference and preserves the integrity of the shared latent representation.

## P. Limitations and Future Work

CLEAR improves concept suppression by learning concept-specific interventions at selected text-encoder layers. This design is effective for precise erasure, but currently requires separate optimization for each target concept. As a result, scaling CLEAR to very large concept libraries may require more efficient parameter sharing, intervention composition, or reusable SAE dictionaries. In addition, our experiments focus on object, identity, nudity, and artist-style concepts. Extending CLEAR to more complex concepts, relational concepts, and policy-dependent safety definitions remains an important direction for future work.

