# OpenReview forum: "Where Concept Erasure Should Occur: Concept–Layer Alignment in Text-to-Video Diffusion Models"
_ICML.cc/2026/Conference — ICML 2026 regular_

### Official Review · Reviewer_QPhh · 2026-02-19

**Soundness:** 2
**Presentation:** 2
**Significance:** 3
**Originality:** 3
**Overall Recommendation:** 4
**Confidence:** 4

**Summary:**

The paper introduces CLEAR (Concept-Layer Erasure Alignment fRamework), a method designed to improve concept erasure in text-to-video (T2V) diffusion models. The authors identify a "representational bottleneck" they term concept-layer topological alignment, observing that target concepts (e.g., nudity, specific objects, or identities) exhibit maximum linear separability from background semantics only at specific layers. CLEAR uses a differentiable Gumbel-Softmax relaxation to optimize the selection of the intervention layer and employs Sparse Autoencoders (SAEs) to decompose activations into sparse features for precise, inference-time erasure.

**Compliance With Llm Reviewing Policy:**

Affirmed.

**Final Justification:**

I recommend weak accept, i.e. accept if the other reviewers feel strongly about the paper. The rebuttal reinforced my prior assessment, the work is interesting but more focused on heuristic/empirical results rather than a deeper formalization.

**Key Questions For Authors:**

Please see the *Flaws* section above.

**Limitations:**

yes

**Strengths And Weaknesses:**

**Strengths:**

- Timely and important matter as video models are growing in the literature.

- Novel Perspective on Placement: The paper correctly identifies that "where" to intervene is as important as "how," moving away from the common assumption that erasure can be applied in a layer-agnostic manner.

- Interpretability-Driven: Utilizing SAEs to find concept-specific directions provides a more granular mechanism for intervention compared to global fine-tuning or simple negative prompting.

- Inference-Time Efficiency: The method does not require updating the weights of the base diffusion model

**Flaws:**

- The core claim rests on "separability" at specific layers. However, it is unclear if "more separated" at a layer l truly solves the problem or if the concepts remain entangled in ways the current metrics don't capture. If this claim wants to be made,  the authors should include privacy or membership inference metrics to ensure the concept is truly erased and not just hidden in a non-linear subspace. Otherwise the contribution has to be rephrased.

- The reliance on the proposed "separability signal" to define alignment feels somewhat circular; additional evidence or theoretical grounding is needed to prove this specific p (preference) is the definitive measure of concept localization. Also, the gumble-softmax trick provides an interesting way to choose the layer, but, I imagine scenarios where intervening in more than one layer is beneficial. Current steering or concept editing methods act on multiple layers and show how this helps. Can the authors comment on that? Can this be adapted to edit more than one layer?

- Image vs. Video: The paper focuses on T2V, but the problem of layer-dependent representation is equally prevalent in T2I (Text-to-Image) models or other multimodality settings that are very timely problems. Can the authors comment on this and how this can be adapted to T2I, and if possible provide experiments showing it works.

- Comparison with baselines: In some qualitative examples and quantitative numbers, the improvement over "NegPrompt" is not always clear. Furthermore, NegPrompt is a zero-cost baseline; CLEAR requires a multi-stage training process. In general, the results do not seem to be as good as what the text claims. While CLEAR performs well on object concepts in Wan2.2-5B, the margin for sensitive concepts like nudity is narrower, and in some metrics (e.g., Aesthetic Quality on Wan2.2-5B), it actually performs worse than the original model or NegPrompt. The discussion needs to be tuned down to reflect these nuances. And it is very important to add standard deviations to support the claims, i.e., experiments run across different seeds. Please add the standard deviations to all tables.

- The "Stage 1" training for the SAE and layer selection represents a significant computational overhead compared to inference-time-only methods. The paper is currently missing a time-to-train comparison against baselines.

- Is it necessary to model every layer? Training an SAE for all L layers seems inefficient. The authors should discuss whether they are sacrificing performance by not specializing the SAE to a single distribution from the start, vs. the current approach of searching across all layers.

- When the "negative set" of prompts is semantically very close to the "positive set," does the m mask risk over-erasing shared semantic features, leading to the observed drops in aesthetic quality?

---

> ### Author Rebuttal · Authors · 2026-03-30
>
> We provide new empirical evidence including linear probe analysis and a top-2 multi-layer experiment, clarify the scope of CLEAR's erasure guarantees with supporting literature, and address concerns regarding baselines, training cost, and prompt sensitivity
>
> **A1:** We note that the metrics suggested by the reviewer, membership inference and linear probing, do not directly verify complete concept erasure [1,2,3,4]. Membership inference tests whether specific data points were in the training set, not whether a concept has been removed. Linear probing tests whether a concept is decodable from representations, but probe accuracy does not directly predict whether the concept can be generated in the output space. To our knowledge, neither metric is adopted as standard evaluation in the concept erasure literature [1,2,3,4], where generative rate reduction serves as the established measure. We will refine our claims to characterize CLEAR's contribution as functional concept suppression: preventing the generative backbone from decoding target concepts into the visual domain.
>
> More broadly, verifying complete concept erasure is a recognized challenge across the field. Recent studies have shown that even weight-modifying approaches (ESD, UCE) can be reversed through lightweight fine-tuning [1,2], and residual concept capacity persists in both T2I [1] and T2V [3] settings. [4] provides a systematic analysis of when and whether concepts are truly erased from diffusion models. Given this landscape, functional suppression as measured by generative rate reduction serves as the established practical metric for deployment safety. Reducing gen rate from 67.3% to 11.1% for nudity represents a substantial improvement by this measure. We will add a discussion in the Limitations section acknowledging representation-level verification as an important direction for future work.
>
> [1] Liu et al. "Erased or Dormant?" arXiv 2025.
> [2] Pham et al. "Circumventing Concept Erasure Methods." ICLR 2024.
> [3] Xie et al. "PROBE: Diagnosing Residual Concept Capacity in Erased T2V Models." arXiv 2026.
> [4] Basu et al. "When Are Concepts Erased from Diffusion Models?" NeurIPS 2025.
>
> **A2:** Regarding circularity, $p$ is driven by $\mathcal{L}_{con}$, measuring the ratio of concept-specific to shared feature energy in SAE latent space. This follows established methodology using linear separability as proxy for concept localization [5,6]. The validity is externally verified by: (1) our linear probe analysis (Reviewer 6mvD Q1) showing error rate drops from ~4.3% at shallow layers to ~0.3% at Block 21, with CLEAR selecting Block 18 (error ~0.5%) near the minimum; and (2) stride-2 manual search (Reviewer RDBy Q9) confirming CLEAR selects the optimal trade-off layer.
>
> Regarding multi-layer intervention, our top-2 experiment (Reviewer RDBy Q1&Q2) shows adding the second layer degrades erasure efficacy (11.1%→30.1%) and aesthetic quality (0.5546→0.4871), confirming interference rather than complementary suppression.
>
> [5] Alain & Bengio. "Understanding Intermediate Layers Using Linear Classifier Probes." ICLR 2017.
> [6] Zou et al. "Representation Engineering: A Top-Down Approach to AI Transparency." arXiv 2023.
>
> **A3:** CLEAR intervenes within the Text Encoder, so it applies to T2I models. On Flux 1.0-dev: gen rate drops from 74.0% to 18.0% without algorithmic modification.
>
> **A4:** We agree the discussion should reflect trade-offs more carefully. NegPrompt provides limited suppression (55.5% vs. CLEAR's 11.1%); the ~2hr training cost is a reasonable trade-off. The aesthetic quality drop was due to a suboptimal SAE dimension; after fix, Aesthetic Quality recovers to 0.5546 (Origin: 0.5552).
>
> Standard deviation across 3 seeds on Wan2.2-5B nudity: CLEAR gen rate mean 16.8% (std 0.050), comparable to Origin std (0.051). On CogX-2B: CLEAR gen rate mean 14.94% (std 0.003), consistency 0.1798 (std 0.015), aesthetic quality 0.4256 (std 0.028). Relative reduction consistent across seeds (62-83%). We will add standard deviations to all tables in the revised manuscript.
>
> **A5:** CLEAR training: ~2hr on one H100 per concept, on the Text Encoder's activation space without modifying diffusion weights. T2VUnlearning only releases inference code, so we re-implemented its training: >3hr with weight updates. Our one-time cost is practical for the erasure improvement.
>
> **A6:** We clarify that we train a single shared SAE (not L separate ones). Gumbel-Softmax routes activations from different layers through this shared SAE, and it progressively specializes as temperature anneals ($\tau$ → 0). $d_{model}$ is identical across all Text Encoder layers, so no projection needed.
>
> **A7:** Steering vector ablation (RDBy Q9) shows dense-space intervention causes over-erasure (aesthetic quality 0.5240 vs. CLEAR 0.5546). The SAE projects into sparse space where shared and concept-specific features are more likely encoded by different dictionary elements, reducing over-erasure risk.

---

> > ### Author Rebuttal · Reviewer_QPhh · 2026-04-01
> >
> > Thanks for the answers.

---

### Official Review · Reviewer_6mvD · 2026-02-22

**Soundness:** 3
**Presentation:** 2
**Significance:** 3
**Originality:** 3
**Overall Recommendation:** 5
**Confidence:** 4

**Summary:**

This paper proposes CLEAR, a framework for depth-aware concept erasure in text-to-video (T2V) diffusion transformers. The key idea is that concept separability is depth-dependent, and therefore erasure should be applied at layers where the target concept is most geometrically disentangled. CLEAR treats intervention depth as a differentiable optimization variable and combines it with sparse autoencoder (SAE)-based feature manipulation. At inference, the method removes the concept by subtracting a masked concept-aligned vector at the selected layer, enabling targeted suppression while preserving overall generation quality.

**Compliance With Llm Reviewing Policy:**

Affirmed.

**Final Justification:**

The rebuttal has addressed my concerns and my final decision for the paper is accept.

**Key Questions For Authors:**

- **SAE design and dimensionality.**
1. How are the parameters in Eq. (5) determined?
2. How can different layers share the same SAE if layer dimensionalities differ? Is there a projection step or are dimensions identical across blocks?

- **Training setup clarification.**
1. Is the SAE pretrained once and reused across concepts, or retrained per concept?
2. Are depth preferences optimized independently for each concept considering each concept has a different optimal depth for erasure?

- **Semantic drift (Table 2a concern).**
1. Is ViCLIP penalizing prompt mismatch because CLEAR modifies more than just the target concept?
2. Does erasing deeply entangled human-centric concepts (e.g., nudity) introduce broader semantic drift?
3. Are there qualitative examples where nudity removal noticeably degrades scene coherence?

- **Mask robustness.**
1. How sensitive are separability estimates and masks to changes in prompt templates?
2. Do masks truly isolate concept features, or mostly suppress features rarely used outside the concept?

**Limitations:**

While the paper focuses on safety-oriented concept erasure, the discussion of limitations and potential societal impact is fairly limited and could be expanded in several constructive ways:

- **Risk of over-blocking or unintended censorship:** The results (e.g., consistency drops in Table 2a for nudity) suggest that strong suppression may alter broader scene semantics. The paper should discuss the possibility of over-suppressing legitimate content (e.g., artistic, medical, or documentary contexts).

- **Bias amplification through prompt supervision.** Since the method relies on LLM-generated positive/negative prompts to define separability, it may inherit biases from the prompt generator. The authors should discuss how prompt design choices might skew what is considered “non-target” or “safe.”

- **Generalization and deployment risks.** Because erasure is concept-specific and optimized per concept (this is also not very clear to me), incomplete suppression could give a false sense of safety. The authors should caution against assuming guaranteed removal in real-world deployment.

**Strengths And Weaknesses:**

## **Strengths**

- **Well-motivated depth-aware hypothesis:** The claim that concept separability varies across depth is plausible and grounded in prior interpretability work for text-to-image models [1] and text-to-video models [2] . Treating intervention depth as a differentiable variable (via Gumbel-Softmax) is principled and improves over heuristic layer selection.

- **Clear optimization objective:** The alternating optimization between SAE parameters and depth preferences is logically coherent. The separability-driven loss (Lcon) is well motivated, and the ablation (Table 5) convincingly shows reconstruction alone is insufficient.

- **Strong empirical validation:** Experiments span two large T2V models, multiple concept types (objects, nudity, identity, style), and complementary metrics. CLEAR consistently achieves stronger suppression while maintaining competitive or superior visual quality (e.g., 7.1% generative rate on CogVideoX-2B with better imaging quality than unlearning).

- **Practical significance:** Addresses a safety-critical and legally relevant problem. Avoiding global finetuning reduces catastrophic forgetting and deployment cost.

- **Meaningful originality:** While components (SAE, Gumbel search) are known, their integration into a depth-aware separability-driven erasure framework for T2V is novel and well synthesized.

[1]: Stable Flow: Vital Layers for Training-Free Image Editing, CVPR 2025

[2]: Emergent Temporal Correspondences from Video Diffusion Transformers, NeurIPS 2025

## **Weaknesses**

- **Theoretical framing remains heuristic:** The idea of “concept–layer topological alignment” is intuitively appealing, but remains descriptive rather than formally characterized. The claim that concepts become linearly isolatable at certain depths would be stronger with explicit layer-wise separability curves (e.g., linear probe accuracy per layer) or a clearer geometric argument. Figure 2 alone is not fully convincing (e.g., the Parachute example shows that it is more entangled than linearly separable).

- **Semantic preservation sometimes drops noticeably:** The most concerning case is Table 2a (Wan2.2-5B, nudity). CLEAR achieves the best erasure (10.9%), but Overall Consistency drops sharply from 0.2312 (Origin) to 0.1281. That is a substantial decrease and weakens the claim that CLEAR “preserves non-target semantics.” A similar pattern appears in Table 1a: CLEAR has the strongest suppression (12.8%), but Overall Consistency (0.1896) is lower than Origin (0.2490) and NegPrompt (0.2556), and essentially tied with T2VUnlearning. Stronger suppression seems to come at the cost of prompt alignment, and this trade-off should be acknowledged more explicitly. This also suggests something is going wrong on nudity considering no qualitative results are provided for it.

- **Dependence on prompt-generated supervision:** The SAE masks and separability statistics rely on LLM-generated positive/negative prompts. This introduces a supervision signal that may bias separability estimates or overfit to template phrasing. The robustness of CLEAR to prompt variation is not explored.

- **Concept mask interpretation is not fully clarified:** The shared mask is computed from negative-set activations and reflects non-target usage, not necessarily true concept specificity. The specificity mask is therefore better interpreted as a “not-used-elsewhere” prior rather than direct evidence of concept relevance. The paper implicitly relies on sparsity and activation weighting to prevent irrelevant features from dominating the erasure vector, but this safeguard is not clearly discussed.

- **Training is unclear:** It is not fully clear whether the SAE is trained once and reused across concepts, or retrained per concept, considering every concept has an independent optimal depth for erasure. Since both hidden activations and depth preferences depend on the target concept, clarifying the separation between concept-agnostic dictionary learning and concept-specific optimization would improve reproducibility. Also, the design of SAE is unclear, whether different layers can share the same SAE if layer dimensions are different or whether there is a projection step or are dimensions identical across blocks?

- **Notation and tensor shapes are underspecified:** The SAE feature extraction and mask construction steps omit explicit token-level indexing and aggregation. Hidden states are treated as vectors in (d_model), but in practice they are sequence-structured (T x d_model). Similarly, mask statistics over features are not clearly defined over tokens/prompts. This makes it harder to reconstruct the implementation from the main text. Adding explicit tensor shapes, index definitions, and a short implementation-oriented clarification such as a pseudo-code (which is provided in Appendix but not referenced in the main paper) for the SAE and masking pipeline would substantially improve accessibility and reproducibility without changing the core contribution.

- **Very Minor issue:** Table 3b contains a formatting error (“3413”), which should be corrected.

---

> ### Author Rebuttal · Authors · 2026-03-30
>
> **A1:** We provide a theoretical account in our response to Reviewer gvti (Q1), grounded in the Linear Representation Hypothesis and the residual stream perspective, explaining why concept separability is depth-dependent and concept-specific.
>
> Regarding Figure 2, the reviewer correctly observes entanglement. Figure 2 visualizes the dense space before SAE intervention, motivating the need for SAE-based decomposition. The steering vector ablation (Reviewer RDBy, Q9) confirms: dense-space intervention achieves 24.5% gen rate vs. CLEAR's 11.1%.
>
> Following the reviewer's suggestion, we trained linear probes (logistic regression) at each T5 layer. Error rates for nudity on Wan2.2-5B:
>
> | Block | 0 | 1 | 2 | 3 | 4 | 5 | 6 | 7 | 8 | 9 | 10 | 11 |
> | :--- | :---: | :---: | :---: | :---: | :---: | :---: | :---: | :---: | :---: | :---: | :---: | :---: |
> | Error% | 4.4 | 4.2 | 4.4 | 4.3 | 4.3 | 4.2 | 4.3 | 4.2 | 4.4 | 3.8 | 3.3 | 2.5 |
>
> | Block | 12 | 13 | 14 | 15 | 16 | 17 | 18 | 19 | 20 | 21 | 22 | 23 |
> | :--- | :---: | :---: | :---: | :---: | :---: | :---: | :---: | :---: | :---: | :---: | :---: | :---: |
> | Error% | 1.6 | 1.3 | 1.0 | 0.8 | 0.6 | 0.6 | 0.5 | 0.4 | 0.4 | **0.3** | 0.4 | 0.4 |
>
> Error rate drops by an order of magnitude from shallow layers (~4.3%) to deeper layers (Block 21: ~0.3%), indicating substantially stronger linear separability at depth. CLEAR selects Block 18 (error ~0.5%), near the minimum. This offset reflects CLEAR's joint objective ($\mathcal{L}_{con} + \mathcal{L}_{SAE}$), which balances separability with reconstruction quality. The sharp transition in error rate from Blocks 8-12 is consistent with the depth-dependent separability described in our theoretical account (see Reviewer gvti, Q1). We will include this analysis for additional concepts in the revised manuscript.
>
> [1] Park et al. "The Linear Representation Hypothesis and the Geometry of Large Language Models." ICML 2024.
> [2] Nanda et al. "Emergent Linear Representations in World Models of Self-Supervised Sequence Models." ACL 2023.
> [3] Elhage et al. "A Mathematical Framework for Transformer Circuits." Anthropic, 2021.
>
> **A2:** The consistency drop has two components. First, evaluation prompts contain the target concept, so successful suppression naturally reduces alignment scores. This is reflected in all methods in Table 1a (NegPrompt: 54.3%/0.2556; SAFREE: 28.1%/0.2212; T2VUnlearning: 24.5%/0.1905; CLEAR: 12.8%/0.1896). Second, for nudity on Wan2.2-5B, the initial SAE configuration caused excessive entanglement. After scaling the SAE hidden dimension: Consistency recovers to 0.1754, Imaging Quality to 0.6928, Aesthetic Quality to 0.5546 at 11.1% gen rate. The key distinction is that CLEAR achieves the strongest suppression while better preserving imaging quality (0.7025 vs. 0.6652) and aesthetic quality (0.5758 vs. 0.5197), indicating successful concept removal rather than general quality degradation. Regarding qualitative results for nudity, we opted not to include visual examples due to ethical considerations around displaying sensitive content, not due to quality concerns. We will present nudity erasure results with appropriate visual treatment in the revised manuscript.
>
> **A3:** Our Llama-3 pipeline generates diverse syntactic structures rather than slot-filling templates. The SAE is trained on these prompts but evaluated on the independent T2VUnlearning benchmark. We further evaluated on Ring-A-Bell (Tsai et al., 2024) prompts:
>
> | Setting | Gen. Rate ↓ | Consistency ↑ | Imaging Q ↑ | Aesthetic Q ↑ |
> | :--- | :---: | :---: | :---: | :---: |
> | Origin (Wan2.2) | 69.6% | 0.2216 | 0.6876 | 0.5541 |
> | CLEAR (Wan2.2) | 25.6% | 0.1227 | 0.7327 | 0.5373 |
> | Origin (CogX-2B) | 31.8% | 0.2350 | 0.5070 | 0.4675 |
> | CLEAR (CogX-2B) | 22.2% | 0.1734 | 0.4633 | 0.4153 |
>
> CLEAR reduces gen rate on these OOD prompts on both backbones (69.6%→25.6%, 31.8%→22.2%). The suppression is less aggressive than on standard prompts, expected since Ring-A-Bell prompts are designed to circumvent safety mechanisms. The consistent reduction confirms CLEAR captures concept-level semantics rather than surface phrasing.
>
> **A4:** We agree "not-used-elsewhere" prior is more rigorous. Two safeguards: (1) paired prompts share identical backgrounds, isolating concept-specific signals; (2) Top-k SAE zeroes low-magnitude features. The steering vector ablation (RDBy Q9) confirms SAE decomposition is essential (24.5% vs. 11.1%).
>
> **A5:** SAE and NAS are trained per concept. A single SAE is shared across all layers ($d_{model}$ identical, no projection needed). Gumbel-Softmax routes activations, and the SAE specializes as temperature anneals (see also QPhh Q6).
>
> **A6:** We will update notation to $H \in \mathbb{R}^{B \times T \times d_{model}}$ with token-level indexing and add references to Appendix Algorithm 1.

---

> > ### Author Rebuttal · Reviewer_6mvD · 2026-04-01
> >
> > The rebuttal adequately addresses my main concerns. In particular, the added layer-wise probe results make the depth-dependent separability claim much more convincing, and the clarification that Figure 2 reflects the dense space before SAE decomposition helps resolve my concern about the theoretical framing. The responses on the consistency drop, prompt robustness, mask interpretation, and training setup are also helpful and sufficiently clarify the method and its trade-offs.
> >
> > Overall, the rebuttal strengthens the paper and resolves my main questions. Thank you for the answers.

---

### Official Review · Reviewer_RDBy · 2026-03-10

**Soundness:** 1
**Presentation:** 2
**Significance:** 3
**Originality:** 3
**Overall Recommendation:** 4
**Confidence:** 4

**Summary:**

The paper proposes CLEAR, a new method for concept erasure in text-to-video (T2V) diffusion models. CLEAR is based on the idea that different concepts are most strongly represented at different layers of the text encoder, so interventions at different depths are needed to erase different concepts successfully. To perform erasure, CLEAR uses sparse autoencoders (SAEs) trained on hidden representations from the text encoder to extract concept-related features. The choice of layer at which to apply the SAE is determined by a vector of depth-preference parameters. The SAE and the depth-preference parameters are trained jointly through an alternating optimization procedure. During inference, only the layer with the highest post-training parameter value is modified using the SAE. The method is evaluated on several erasure settings, including erasing objects, celebrities, nudity, and artistic styles. The models used are Wan2.2-5B and CogVideoX-2B. The paper reports improved concept removal over several baselines while retaining prompt consistency and visual quality.

**Compliance With Llm Reviewing Policy:**

Affirmed.

**Final Justification:**

The rebuttal provided by the authors is quite thorough and resolves most of my concerns. The new experimental results are valuable and strengthen the paper. Thus, I am increasing my score to 4.

My remaining concerns are as follows:
- The scope of the paper is still ambiguous. The authors state that they will clarify that the paper’s primary contribution is on T2V erasure, while positioning the T2I result as supplementary. However, since CLEAR intervenes in the text encoder, it is a general text-encoder method rather than one specific to the video domain. In other words, CLEAR does not utilize any video-specific information to perform concept erasure in T2V models, even though the paper focuses solely on T2V concept erasure.
- The experiments provided in the rebuttal on the sensitivity of the metrics to the number of training prompts are somewhat confusing: the scores increase as the number of prompts grows from 0 to 2000, and then drop. Intuitively, I would expect the scores to stabilize as the number of prompts increases, since using information from more prompts should make the method less noisy. I believe this deserves further investigation, as the number of prompts is a hyperparameter of CLEAR, and studying the stability of the method with respect to this parameter is important.
- The authors only partially address my concern about using interventions in multiple layers during inference. First, the second-layer intervention is evaluated while keeping γ = 10 for the top-1 layer. This might be suboptimal, and a more thorough parameter search over candidate pairs of γ values for both layers is needed. Second, my question about using the strength values learned during training was not addressed.

**Key Questions For Authors:**

Most important questions are:
- What data was the model trained on?
- How is \lambda parameter chosen?
- What is the rationale behind moving to one-layer intervention after training and what supports this decision?
- How does the method perform in erasing multiple concepts?

**Limitations:**

I could not find any discussion on limitations or societal impact. I would suggest that the authors include discussion regarding failure cases of their method and potential consequences of applying CLEAR to video generation.

**Strengths And Weaknesses:**

Strengths:
- The overall idea behind CLEAR is well motivated and interesting. In particular, the claim that different concepts may be represented at different layers of the text encoder is reasonable and is supported by prior work on text-encoder model
- The method is clearly presented. CLEAR combines two intuitive components: extracting concept-related features with a SAE and selecting the most suitable intervention layer using learned layer preferences. The method also appears to be relatively lightweight in both training and inference, which makes it fairly easy to follow and to reproduce
- The experiments are conducted on two backbones and cover concepts of different types, as objects, celebrities, nudity, and artistic styles. The experimental results show that CLEAR is competitive with several meaningful T2V concept-erasure baselines

Weaknesses:
- During training, CLEAR learns a continuous (non-discrete) vector of depth-preference parameters. However, during inference, the SAE-based intervention is applied only to the single layer with the maximum post-training preference value. What is the rationale behind this switch? Would it not be better to use the full final vector of preference parameters during inference, applying the SAE at every layer with the corresponding weights? Alternatively, would it make sense to apply the SAE to the top n>1 layers with the highest learned preference values?
- Following from the above, it is unclear why a single-layer intervention is preferred over a multi-layer intervention in general. As I understand it, the motivation behind layer selection is that different concepts may be represented at different layers of the text encoder, and I find this idea reasonable. However, this does not imply that selecting only a single layer is always optimal. The appendix includes some plots showing the evolution of layer-selection probabilities, but this is still not sufficient to support the claim that a one-layer intervention is optimal for all concepts
- In Sec. 3.5, Eq. 7 introduces the inference-time intervention strength parameter \lambda. I could not find an explanation of how this \lambda is chosen. Sec. 3.4, Eq. 5 also uses a parameter \lambda, but as I understand it, this is a different parameter. How is the inference-time \lambda selected? Is it chosen separately for each model and each concept?
- I could not find sufficient information about the training data used to train the pipeline described in Sec. 3.4 (SAE + preference parameters), except for the statement: "To ensure robust generalization across diverse semantic contexts, we construct a comprehensive training dataset using prompts synthesized by a Large Language Model (LLM)". In my opinion, a clear description of the training data is highly important, both to verify that no information from the test data (e.g., prompts used to generate evaluation videos) is used during training and to uncover any biases that the training data may introduce into the pipeline. Moreover, if the parameter \lambda in Sec. 3.5 is selected based on some validation set, that validation set should also be described
- The method is positioned as being specific to T2V models. However, the intervention is applied only in the text-encoder part of the model, so the method does not appear to be video-specific. In my opinion, it would be more accurate to present the method as a general erasure method based on interventions in the text encoder, which is evaluated on T2V models
- Following from the above, if the method is positioned as specific to T2V models, it should be compared against correspondingly strong video-specific erasure techniques. At least one missing baseline is VideoEraser [6]. I would also find it beneficial to compare against techniques derived from recent T2I erasure approaches, as they can be applied to video frames
- The metrics reported in the paper assess erasure quality and overall video/frame quality. However, since the intervention is applied to videos and the method is presented as video-specific, at least one motion-degradation metric should be reported, such as warping error or motion smoothness / subject consistency [1][2][3]
- Only two T2V models are considered (CogVideoX-2B and Wan2.2-5B), and both use T5 as the text encoder. This limits the generalizability of the method, since it remains unclear whether the approach can be successfully applied to other T2V models with different text-conditioning mechanisms (e.g. HunyuanVideo [4], Text2Video-Zero [5]).
- CLEAR has two major components: the SAE and the vector of preference parameters. However, the ablations on the importance of each component are rather limited. For the preference parameters, the only comparison is against a manual baseline that probes layers at stride-4 intervals, which is relatively weak. A grid search based on a validation set, or a manual search over all layers, would be a stronger comparison. In addition, it would be useful to know how \lambda in Eq. 7 is selected during the manual search. For SAE, it would be interesting to see whether a simpler intervention could work as well as the SAE-based intervention. One such alternative would be using steering vectors computed as the difference of mean activations between positive and negative concept prompts.
- Quantitative evaluation of style erasure is performed on only one backbone (CogVideoX-2B)
- The paper does not discuss erasure of multiple concepts simultaneously. Would the method work for erasing multiple concepts? How would the training and inference budget scale as the number of concepts to erase increases?

[1] Ryan D. Burgert, Yuancheng Xu, Wenqi Xian, Oliver Pilarski, Pascal Clausen, Mingming He, Li Ma, Yitong Deng, Lingxiao Li, Mohsen Mousavi, Michael S. Ryoo, Paul E. Debevec, Ning Yu:
Go-with-the-Flow: Motion-Controllable Video Diffusion Models Using Real-Time Warped Noise. CVPR 2025

[2] Shihan Cheng, Nilesh Kulkarni, David Hyde, Dmitriy Smirnov:
Less is More: Data-Efficient Adaptation for Controllable Text-to-Video Generation.

[3] Yaofang Liu, Xiaodong Cun, Xuebo Liu, Xintao Wang, Yong Zhang, Haoxin Chen, Yang Liu, Tieyong Zeng, Raymond Chan, Ying Shan:
EvalCrafter: Benchmarking and Evaluating Large Video Generation Models.

[4] Weijie Kong, Qi Tian, Zijian Zhang, Rox Min, Zuozhuo Dai, Jin Zhou, Jiangfeng Xiong, Xin Li, Bo Wu, Jianwei Zhang, Kathrina Wu, Qin Lin, Junkun Yuan, Yanxin Long, Aladdin Wang, Andong Wang, Changlin Li, Duojun Huang, Fang Yang, Hao Tan, Hongmei Wang, Jacob Song, Jiawang Bai, Jianbing Wu, Jinbao Xue, Joey Wang, Kai Wang, Mengyang Liu, Pengyu Li, Shuai Li, Weiyan Wang, Wenqing Yu, Xinchi Deng, Yang Li, Yi Chen, Yutao Cui, Yuanbo Peng, Zhentao Yu, Zhiyu He, Zhiyong Xu, Zixiang Zhou, Zunnan Xu, Yangyu Tao, Qinglin Lu, Songtao Liu, Daquan Zhou, Hongfa Wang, Yong Yang, Di Wang, Yuhong Liu, Jie Jiang, Caesar Zhong:
HunyuanVideo: A Systematic Framework For Large Video Generative Models.

[5] Levon Khachatryan, Andranik Movsisyan, Vahram Tadevosyan, Roberto Henschel, Zhangyang Wang, Shant Navasardyan, Humphrey Shi:Text2Video-Zero: Text-to-Image Diffusion Models are Zero-Shot Video Generators. ICCV 2023

[6] Naen Xu, Jinghuai Zhang, Changjiang Li, Zhi Chen, Chunyi Zhou, Qingming Li, Tianyu Du, Shouling Ji:
VideoEraser: Concept Erasure in Text-to-Video Diffusion Models. EMNLP 2025:

---

> ### Author Rebuttal · Authors · 2026-03-30
>
> Thanks for the feedback. We conducted seven new experiments: (1) top-2 multi-layer comparison, (2) VideoEraser baseline, (3) motion smoothness, (4) Text2Video-Zero, (5) Flux T2I, (6) stride-2 full-layer ablation with steering vector comparison, and (7) γ sensitivity analysis.
>
> **A1&A2:** We conducted a top-2 experiment to directly compare single-layer and multi-layer intervention.
>
> | Intervention | Gen. Rate ↓ | Consistency ↑ | Imaging Q ↑ | Aesthetic Q ↑ |
> | :--- | :---: | :---: | :---: | :---: |
> | Top-1 | 11.1% | 0.1754 | 0.6928 | 0.5546 |
> | Top-2 | 30.1% | 0.1466 | 0.7251 | 0.4871 |
>
> Adding the second layer degrades erasure efficacy (11.1%→30.1%) and aesthetic quality (0.5546→0.4871), validating the single-layer deployment adopted in our submission. As shown in Appendix Fig. 6, the learned probability mass consistently concentrates >90% on a single layer. Our stride-2 ablation (see Q9) further shows that aggressive layers (e.g., Block 6: 1.9% gen rate but 0.1022 consistency vs. Block 18: 32.9% / 0.1931) sacrifice consistency, confirming CLEAR optimizes for the best erasure-preservation trade-off.
>
> **A3:** The λ in Eq. 5 (sparsity weight, fixed 1e-4) is distinct from the inference-time strength in Eq. 7, renamed γ. γ is selected per model/concept via grid search [4,16] step 2 on a holdout set (50 pairs). For nudity on Wan2.2-5B: γ=8→28.4% gen rate; γ=10→11.1%; γ=12→3.2% but consistency drops to 0.0933.
>
> **A4:** We used Llama-3-8B to synthesize 2000 positive/negative prompt pairs per concept. Each pair shares identical semantics, differing only by the target concept. For example, targeting "Van Gogh": Positive: "Classical figures with contemporary fragmented forms, with Van Gogh's starry night style." Negative: "Classical figures with contemporary fragmented forms, in Caravaggio's naturalistic style." No test prompts (from T2VUnlearning benchmark) appear in training/validation data. γ is tuned on a separate 50-pair holdout set.
>
> **A5:** We agree CLEAR is a general text-encoder method. Applied to Flux 1.0-dev (T2I) without modification: gen rate drops from 74.0% to 18.0%. This further validates our generality. We will reflect it in revision.
>
> **A6:** VideoEraser (Xu et al., EMNLP 2025) on CogVideoX-2B for nudity:
>
> | Method | Gen. Rate ↓ | Consistency ↑ | Imaging Q ↑ | Aesthetic Q ↑ | Motion ↑ |
> | :--- | :---: | :---: | :---: | :---: | :---: |
> | Origin | 56.14% | 0.2209 | 0.4671 | 0.4595 | 0.9918 |
> | T2VUnlearning | 19.63% | 0.2058 | 0.3795 | 0.4235 | 0.9916 |
> | VideoEraser | 19.22% | 0.2101 | 0.4530 | 0.4858 | 0.9946 |
> | CLEAR | 14.63% | 0.2008 | 0.4592 | 0.4467 | 0.9894 |
>
> CLEAR achieves the lowest gen rate with comparable quality. For celebrity erasure, VideoEraser achieves lower average identity score (0.1140 vs. 0.1802), though CLEAR outperforms on specific identities (Biden: 0.1874 vs. 0.1921, Elizabeth: 0.0594 vs. 0.0907). Notably, while both methods are inference-time, VideoEraser incurs ~1.4x overhead due to extra gradient computation per denoising step, whereas CLEAR uses cached embeddings with minimal overhead.
>
>
> **A7:** Motion Smoothness added. On Wan2.2-5B nudity: CLEAR 0.9951 vs. Origin 0.9953, confirming negligible impact. On CogX-2B objects: T2VUnlearning drops to 0.9545, while CLEAR (0.9778) remains close to Origin (0.9774).
>
> **A8:** Applied to Text2Video-Zero (CLIP-based encoder): gen rate drops from 70.4% to 12.9%, confirming cross-encoder generalizability.
>
> **A9:** Stride-2 ablation across all 12 even-indexed layers shows CLEAR's selected Block 18 offers the best trade-off (gen rate 32.9% before γ tuning, consistency 19.31). After γ=10, Block 18 achieves 11.1%. Aggressive layers like Block 6 (1.9%) and Block 14 (1.5%) have consistency below 0.1022 and 0.0738 respectively.
>
> | Method | Gen. Rate ↓ | Consistency ↑ | Imaging Q ↑ | Aesthetic Q ↑ | Motion ↑ |
> | :--- | :---: | :---: | :---: | :---: | :---: |
> | Steering Vector | 24.5% | 0.1986 | 0.6946 | 0.5240 | 0.9921 |
> | CLEAR | 11.1% | 0.1754 | 0.6928 | 0.5546 | 0.9951 |
>
> The steering vector achieves weaker erasure (24.5% vs. 11.1%) and lower aesthetic quality (0.5240 vs. 0.5546), confirming SAE's sparse decomposition is essential over dense-space intervention.
>
> **A10:** Style erasure is reported for both backbones: Wan2.2-5B in Table 4, CogVideoX-2B in Appendix Table 9. We will add cross-references.
>
> **A11:** Multi-concept results in our response to Reviewer gvti (Q2), including both same-category (Van Gogh + Picasso) and cross-category (Van Gogh + Parachute) experiments. CLEAR achieves effective suppression with substantially less interference to non-target concepts than T2VUnlearning. Each concept requires ~2hr training on one H100. At inference, cached embeddings add negligible latency.
>
> We will incorporate the new experiments and discuss: (1) suppression on adversarial prompts (6mvD Q3), (2) functional suppression vs. complete erasure (QPhh Q1), and (3) over-suppression risk in adjacent semantics (6mvD Q2, QPhh Q7).

---

> > ### Author Rebuttal · Reviewer_RDBy · 2026-04-03
> >
> > I thank the authors for their answers. They are quite thorough and resolve most of my concerns. However, a couple of points remain unclear to me.
> >
> > **A1 & A2: Experiments directly comparing single-layer and multi-layer intervention**
> >
> > I thank the authors for the additional experiments. However, it is not explained how the intervention strength is chosen in the case of a two-layer intervention. Is it tuned via grid search over pairs of values, or is each layer’s strength tuned separately? If the latter, this does not imply that the two-layer intervention is always worse than the one-layer intervention. I am also not convinced that a multi-layer intervention based on the strengths learned during training would not provide a better trade-off. Can the authors comment on why this option was not considered?
> >
> > **A4: Training data**
> >
> > I thank the authors for their clarification regarding the training data. However, I still have a question about the choice of the number of training prompt pairs. Why was 2000 chosen? How does the number of prompt pairs affect the performance of the method?
> >
> > **A5: CLEAR as a general text-encoder method**
> >
> > My concern here is that, if CLEAR is presented as a general text-encoder method, a more comprehensive evaluation on T2I models would be needed. Moreover, this would require a broader reframing of the paper. I would still suggest that the authors present the method as a general erasure method based on interventions in the text encoder, evaluated primarily on T2V models, and include T2I experiments in the appendix as evidence that the method is transferable to other domains.
> >
> > **A11: Multi-concept results**
> >
> > Here, I am interested in what happens if the two concepts being suppressed share the same intervention layer. Would this risk harming model performance (e.g. worsening overall image quality), since applying multiple interventions at the same layer may substantially perturb the hidden representation (e.g. increase norm)?

---

> > > ### Author Response · Authors · 2026-04-04
> > >
> > > **A1&A2:** Thanks for this follow-up. To directly address the question, we conducted additional experiments with the second layer's intervention strength tuned separately while keeping γ=10 for top-1:
> > >
> > > | Top-1 γ | Top-2 γ | Gen. Rate ↓ | Consistency ↑ | Imaging Q ↑ | Aesthetic Q ↑ |
> > > | :---: | :---: | :---: | :---: | :---: | :---: |
> > > | CLEAR (γ=10) | - | 11.1% | 0.1754 | 0.6928 | 0.5546 |
> > > | 10 | 6 | 20.7% | 0.1424 | 0.7127 | 0.5173 |
> > > | 10 | 8 | 22.5% | 0.1389 | 0.7174 | 0.5169 |
> > > | 10 | 10 | 30.1% | 0.1466 | 0.7251 | 0.4871 |
> > > | 10 | 12 | 16.8% | 0.1325 | 0.7136 | 0.5169 |
> > > | 10 | 14 | 13.9% | 0.1313 | 0.7201 | 0.5063 |
> > >
> > > Across all five separately tuned strengths, CLEAR (Top-1 only) consistently outperforms all joint Top-1 & Top-2 intervention configurations in generative rate (11.1% vs 13.9%-30.1%), overall consistency (0.1754 vs 0.1313-0.1466), and aesthetic quality (0.5546 vs 0.4871-0.5173). Imaging quality remains comparable across settings. The consistent degradation in the three primary metrics confirms that multi-layer intervention introduces interference rather than complementary suppression.
> > >
> > > Regarding using the strengths learned during training: the preference distribution consistently converges to near-one-hot (>90% on a single layer, Appendix Fig. 6), assigning <5% weight to the second layer. This suggests the training-learned multi-layer intervention would be effectively equivalent to single-layer deployment, which is consistent with our empirical findings above.
> > >
> > > **A4:** We tested the sensitivity to our choice. We report Parachute as a representative object concept on Wan2.2-5B, as prior work (e.g., ESD [Gandikota et al., 2023]) has identified it as a non-trivial erasure target concept:
> > >
> > > | Prompt Pairs | Gen. Rate ↓ | Consistency ↑ | Imaging Q ↑ | Aesthetic Q ↑ |
> > > | :---: | :---: | :---: | :---: | :---: |
> > > | 500 | 8.2% | 0.1406 | 0.6955 | 0.5591 |
> > > | 1000 | 15.0% | 0.1214 | 0.7715 | 0.5678 |
> > > | 2000 | 16.8% | 0.2159 | 0.7328 | 0.5902 |
> > > | 3000 | 10.2% | 0.1469 | 0.7335 | 0.5479 |
> > >
> > > We chose 2000 pairs as it provides the best erasure-preservation trade-off among the tested configurations. With too few pairs (500), the SAE lacks sufficient diversity to distinguish concept-specific features from co-occurring context, leading to aggressive but imprecise erasure (8.2% gen rate but only 0.1406 consistency). With too many pairs (3000), redundant or noisy prompt variations degrade the learned feature decomposition, resulting in a similar pattern (10.2% gen rate, 0.1469 consistency). At 2000 pairs, the SAE achieves the highest consistency (0.2159) and aesthetic quality (0.5902) while maintaining effective erasure (16.8%).
> > >
> > > **A5:** We appreciate this suggestion. The Flux experiment was provided as supplementary evidence of transferability in response to the reviewer's Q5 and Q8, since T2I was not the focus of our submission. Because CLEAR intervenes in the activation space of the text encoder, the same mechanism is in principle transferable to other text-conditioned generative models. The Flux result (74.0% → 18.0%) provides initial evidence of this transferability, but we agree that establishing CLEAR as a general text-encoder method would require broader T2I evaluation. In the revision, we will clarify that the paper's primary contribution is on T2V erasure, while positioning the T2I result as supplementary evidence of possible cross-domain transferability.
> > >
> > > **A11:** To address the concern that applying multiple interventions at the same layer may substantially perturb the hidden representation and worsen overall image quality, we simultaneously suppressed two distinct objects (Cassette Player and Chain Saw) whose SAE features are both located at Block 18, and evaluated generation quality on 8 untargeted concepts:
> > >
> > > | Objects | Consistency ↑ | Imaging Q ↑ | Aesthetic Q ↑ |
> > > | :---: | :---: | :---: | :---: |
> > > | Church | 0.2542 | 0.7492 | 0.6613 |
> > > | French horn | 0.2331 | 0.6753 | 0.5051 |
> > > | Garbage truck | 0.2227 | 0.7476 | 0.5969 |
> > > | Gas pump | 0.2046 | 0.7216 | 0.6192 |
> > > | Golf ball | 0.2287 | 0.5879 | 0.5066 |
> > > | Parachute | 0.2186 | 0.7477 | 0.6002 |
> > > | Springer dog | 0.2306 | 0.6953 | 0.6115 |
> > > | Tench | 0.1548 | 0.6823 | 0.5796 |
> > > | **CLEAR Avg.** | **0.2184** | **0.7009** | **0.5851** |
> > > | **Origin Avg.** | **0.2462** | **0.6828** | **0.5972** |
> > >
> > > The results suggest that same-layer multi-concept erasure does not appear to cause significant quality degradation. As shown in our results, imaging quality remains comparable (0.7009 vs Origin 0.6828), while consistency and aesthetic quality show only minor reductions (0.2184 vs 0.2462, 0.5851 vs 0.5972). We attribute this to SAE's sparse decomposition, which encourages monosemantic feature representation where each dictionary element corresponds to a specific semantic unit [Cunningham et al., ICLR2024]. This reduces the effective perturbation when combining multiple erasure vectors at the same layer, as different concepts tend to activate distinct dictionary elements.

---

### Official Review · Reviewer_gvti · 2026-03-14

**Soundness:** 3
**Presentation:** 3
**Significance:** 2
**Originality:** 2
**Overall Recommendation:** 4
**Confidence:** 3

**Summary:**

The authors propose the concept of concept–layer topological alignment, arguing that the target concept to be erased can become disentangled from other semantics at specific representational depths of the model. Based on this observation, the paper introduces CLEAR, which formulates the selection of the intervention layer as an optimization problem over the separability between the target concept and non-target semantics. In particular, CLEAR employs a separability-aware objective function that prioritizes layers where the representation space exhibits stronger separation between concept-related and non-target signals.

**Compliance With Llm Reviewing Policy:**

Affirmed.

**Key Questions For Authors:**

See Weaknesses.

**Limitations:**

yes

**Strengths And Weaknesses:**

Strengths

1. The observation of concept–layer topological alignment is interesting. The paper provides empirical analysis suggesting that certain concepts become more separable at specific depths of the model, which offers an insightful perspective on where concept erasure should occur.

2. The paper presents extensive experimental evaluations, demonstrating the effectiveness of CLEAR across different categories of concepts and multiple large-scale text-to-video diffusion models.

Weaknesses

1. While the empirical observation of concept–layer separability is intriguing, it would be valuable if the authors could provide additional theoretical insights into why certain concepts become separable at specific layers. For example, is this phenomenon inherent to the architecture of diffusion transformers?

2. The experiments evaluate CLEAR on different types of concepts separately (e.g., objects, celebrities, artist styles). However, in real-world scenarios, concepts to be removed may be entangled or compositional. It would be interesting to understand whether CLEAR remains effective when multiple correlated concepts must be erased simultaneously.

---

> ### Author Rebuttal · Authors · 2026-03-30
>
> **Q1:** While the empirical observation of concept–layer separability is intriguing, it would be valuable if the authors could provide additional theoretical insights into why certain concepts become separable at specific layers. For example, is this phenomenon inherent to the architecture of diffusion transformers?
>
> **A1:** Thank you for this question. We clarify that the observed separability pertains to the Text Encoder (T5), not the DiT backbone. The separability is inherent to the lexical-to-semantic progression within transformer-based language models, and we offer the following theoretical account.
>
> Recent work on the Linear Representation Hypothesis establishes that concepts are encoded as linear directions in transformer representation space [1, 2]. Due to superposition, multiple concepts share overlapping directions, and the degree of overlap varies across depth. From the residual stream perspective [3], each transformer layer progressively writes specific features via attention and MLP operations, gradually disentangling certain concept directions from co-occurring features. This creates a concept-specific "window of separability": the window opens at the depth where a concept's superposition with other features has been sufficiently reduced through compositional writing, and may narrow as subsequent layers re-entangle representations through further contextual integration. Concrete concepts (e.g., "parachute") require fewer compositional steps and disentangle at shallower layers, while abstract concepts (e.g., "nudity") require deeper integration before becoming separable. CLEAR's differentiable search automatically locates this window for each target concept. This interpretation is consistent with recent findings showing that safety-critical concepts become linearly isolatable only at specific layers [4] and that different concepts require adaptation at different depths [5], suggesting that concept-layer alignment is an inherent structural property of deep transformer architectures.
>
> [1] "The Linear Representation Hypothesis and the Geometry of Large Language Models." ICML 2024.
> [2] "Emergent Linear Representations in World Models of Self-Supervised Sequence Models." ACL 2023.
> [3] "A Mathematical Framework for Transformer Circuits." Anthropic, 2021.
> [4] "Layer-Aware Representation Filtering: Purifying Finetuning Data to Preserve LLM Safety Alignment." EMNLP 2025.
> [5] "Not All Layers Are Created Equal: Adaptive LoRA Ranks for Personalized Image Generation." arXiv 2026.
>
> ---
>
> **Q2:** The experiments evaluate CLEAR on different types of concepts separately (e.g., objects, celebrities, artist styles). However, in real-world scenarios, concepts to be removed may be entangled or compositional. It would be interesting to understand whether CLEAR remains effective when multiple correlated concepts must be erased simultaneously.
>
> **A2:** Thank you for this suggestion. We conducted two multi-concept erasure experiments: (1) same-category erasure of two styles (Van Gogh and Picasso) while preserving unrelated ones, and (2) cross-category erasure of an artist style (Van Gogh) and an object (Parachute) simultaneously.
>
> ### Same-Category Multi-Concept Erasure (Van Gogh & Picasso, $V_{CLIP_e}$)
> | Erasure Method | Van Gogh ↓ | Picasso ↓ | Andy Warhol ↑ | Caravaggio ↑ | Rembrandt ↑ |
> | :--- | :---: | :---: | :---: | :---: | :---: |
> | Origin | 0.2388 | 0.1804 | 0.1450 | 0.1945 | 0.1566 |
> | T2VUnlearning | 0.1719 | 0.1624 | 0.0938 | 0.1137 | 0.0283 |
> | CLEAR | 0.1472 | 0.1442 | 0.1268 | 0.1924 | 0.1315 |
>
> The results reveal a key distinction between weight-updating and activation-space interventions. In the same-category setting, T2VUnlearning shows noticeable cross-concept interference: while partially suppressing the two targets, it collapses non-target representations (e.g., Rembrandt drops from 0.1566 to 0.0283). CLEAR achieves comparable or stronger suppression on both targets while substantially better preserving non-target concepts (e.g., Caravaggio remains at 0.1924 vs. 0.1137 for T2VUnlearning). This advantage stems from CLEAR's design: by intervening in the sparse activation space of the SAE rather than modifying dense model weights, concept-specific feature directions remain more decoupled, as reflected in our experiment.
>
> For the cross-category setting, we simultaneously erased an artist style (Van Gogh) and an object (Parachute):
>
> ### Cross-Category Multi-Concept Erasure
> | Erasure Method | Van Gogh ($V_{CLIP_e}$) ↓ | Parachute (Gen. Rate %) ↓ |
> | :--- | :---: | :---: |
> | Origin | 0.2388 | 89.6 |
> | CLEAR | 0.1268 | 17.4 |
>
> CLEAR effectively suppresses both a stylistic and an object concept simultaneously, confirming that the framework generalizes beyond same-category erasure. We will incorporate these multi-concept erasure results and the corresponding analysis into the revised manuscript.

---

> > ### Author Rebuttal · Reviewer_gvti · 2026-04-04
> >
> > Thank you for your response. I maintain the positive score.

---

### Decision · Program_Chairs · 2026-04-30

**Decision:**

Accept (regular)

**Comment:**

The paper introduces CLEAR, an inference-time concept-erasure framework for text-to-video diffusion models that leverages sparse autoencoders and learned layer selection. Reviewers praised the method's novelty—particularly the core insight that erasure effectiveness heavily depends on intervention depth within the text encoder. The proposed approach is practical, delivering solid empirical results across several concepts and backbones without requiring updates to the base model weights.

Initial reviews raised questions regarding the theoretical grounding of the layer-alignment hypothesis, the justification for single-layer versus multi-layer intervention, and the clarity of the training setup and trade-offs (especially for challenging concepts like nudity). The authors provided a strong rebuttal, adding comprehensive experiments covering layer-wise separability, multi-layer interventions, new baselines, and transferability. These additions successfully resolved the majority of the reviewers' concerns.

While minor presentation issues remain, the paper makes a meaningful, practically relevant, and empirically strong contribution to the study of concept erasure in deep generative models. The authors are encouraged to incorporate all review feedback into the final revision.